# Thinking inside the box: Restoring the propolis envelope facilitates honey bee social immunity

**Maggie Shanahan**[1]☉*, **Michael Simone-Finstrom**[2]☉*, **Philip Tokarz**[2], **Frank Rinkevich**[2], **Quentin D. Read**[3], **Marla Spivak**[1]

**1** Department of Entomology, University of Minnesota, Saint Paul, Minnesota, United States of America, **2** Honey Bee Breeding, Genetics and Physiology Research Laboratory, United States Department of Agriculture—Agricultural Research Service, Baton Rouge, Louisiana, United States of America, **3** United States Department of Agriculture—Agricultural Research Service Southeast Area, Raleigh, North Carolina, United States of America

☉ These authors contributed equally to this work.
* mshanaha@umn.edu (MS); michael.simonefinstrom@usda.gov (MSF)

**Data Availability Statement:** All relevant data are within the paper and its Supporting Information files.

## Abstract

When wild honey bee colonies (*Apis mellifera*) nest in hollow tree cavities, they coat the rough cavity walls with a continuous layer of propolis, a substance comprised primarily of plant resins. Studies have shown that the resulting "propolis envelope" leads to both individual- and colony-level health benefits. Unfortunately, the smooth wooden boxes most commonly used in beekeeping do little to stimulate propolis collection. As a result, most managed bees live in hives that are propolis-poor. In this study, we assessed different surface texture treatments (rough wood boxes, boxes outfitted with propolis traps, and standard, smooth wood boxes) in terms of their ability to stimulate propolis collection, and we examined the effect of propolis on colony health, pathogen loads, immune gene expression, bacterial gene expression, survivorship, and honey production in both stationary and migratory beekeeping contexts. We found that rough wood boxes are the most effective box type for stimulating propolis deposition. Although the use of rough wood boxes did not improve colony survivorship overall, *Melissococcus plutonius* detections via gene expression were significantly lower in rough wood boxes, and viral loads for multiple viruses tended to decrease as propolis deposition increased. By the end of year one, honey bee populations in migratory rough box colonies were also significantly larger than those in migratory control colonies. The use of rough wood boxes did correspond with decreased honey production in year one migratory colonies but had no effect during year two. Finally, in both stationary and migratory operations, propolis deposition was correlated with a seasonal decrease and/or stabilization in the expression of multiple immune and bacterial genes, suggesting that propolis-rich environments contribute to hive homeostasis. These findings provide support for the practical implementation of rough box hives as a means to enhance propolis collection and colony health in multiple beekeeping contexts.

**Funding:** M. Spivak and M.S-F. received funding from the United States Department of Agriculture - National Institute of Food and Agriculture (https://www.nifa.usda.gov; grant number 2018-67013-27532). M.S-F. and F.R. received funding from Project Apis m. (https://www.projectapism.org) and the United States Department of Agriculture - Agricultural Research Service (https://www.ars.usda.gov; project number 6050-21000-016-000-D). The funders had no role in study design, data collection and analysis, decision to publish, or preparation of the manuscript.

**Competing interests:** The authors have declared that no competing interests exist.

# Introduction

Although many beekeeping practices are designed to support colony health, some inadvertently constrain the natural defenses (or mechanisms of social immunity) that help honey bees (*Apis mellifera* L.) thrive in an unmanaged context [1–3]. When external conditions are favorable (i.e., when colonies have access to abundant floral resources and are exposed to few external stressors), constraining these defenses may not significantly impact colony health. However, many honey bee colonies face conditions that are far from favorable [4, 5]. In the U.S. and around the world, industrial agriculture increases bees' exposure to agrochemicals [6] and pathogens [1, 7], and limits access to diverse forage resources [8, 9], leading to high levels of colony loss (reviewed by Shanahan [10]). These stressors impact both large-scale, migratory beekeeping operations–where colonies providing pollination services participate directly in industrial agriculture–and stationary, small-scale apiaries, which may interface with industrial agriculture less directly [11]. While restoring honey bees' natural defenses will not address the full spectrum of stressors that currently cause colony loss, recovering these health-supportive behaviors could represent one valuable step towards improved honey bee health [12]. Propolis collection is one example of a natural defense that could be integrated by beekeepers working at a variety of scales to improve colony health.

Honey bees collect antimicrobial resins produced by plants [13, 14], and mix this material with beeswax to make propolis, which serves multiple purposes inside the hive [15]. When wild honey bee colonies nest in hollow tree cavities, the cracks and crevices found inside the tree stimulate bees to lay down a continuous layer of propolis, called the "propolis envelope" [16, 17]. However, the smooth wood boxes that most beekeepers use have few cracks and crevices and do little to stimulate propolis collection [18]. Moreover, since propolis gums up beekeeping equipment, propolis collection has long been considered a sticky inconvenience, and over time beekeepers have selected against propolis collection traits, particularly in the U.S. [15]. As a result, most managed bees live in hives that are propolis-poor. This is concerning because a growing body of evidence suggests that propolis is an important part of a colony's social immunity and could reduce the impact of some of the stressors that threaten honey bee health both within and beyond industrialized agricultural landscapes [19].

Propolis-rich environments have been shown to support honey bee colony health in a variety of ways (reviewed by Simone-Finstrom and Spivak [15], Simone-Finstrom et al. [19]). In addition to modulating immune gene expression and improving colony strength and survivorship, propolis may help mitigate pathogen impacts [18–20]. One study demonstrated that honey bee colonies increased resin-foraging when infected with the fungal parasite *Ascosphaera apis* (Maasen ex Claussen), and chalkbrood infection was reduced in hives painted with a propolis extract solution [21]. In another study, when propolis extract was applied to larval rearing cells in amounts similar to those found in brood comb, the survival and reproduction of *Varroa* mites (*Varroa destructor*, Anderson and Trueman) was decreased compared to propolis-free controls [22]. The colony-level implications of this effect are unclear. When propolis was added to one set of colonies and removed from another to create propolis-rich and propolis-poor hive environments, no significant differences in mite infestation were observed, though propolis did appear to interfere with the transmission of deformed wing virus (DWV), which could have important implications for colony health [20]. Lastly, propolis may help mitigate *Nosema ceranae* Fries infection (*Vairimorpha ceranae* [23]), as bees fed with a propolis extract had significantly reduced *V. ceranae* spore loads [24–26]. Though honey bees are not known to consume propolis directly, honey does contain numerous propolis-derived compounds, and these may help protect bees against pathogens, toxins, and other important stressors [27, 28].

There is also evidence that propolis has a stabilizing effect on the honey bee microbiome. Multiple studies have shown that bees from propolis-rich environments (i.e., hives whose surface textures are modified to encourage propolis collection) tend to have more consistent (i.e., less diversity, lower abundance) microbial communities, and bees from propolis-poor environments tend to host a greater diversity of microbiota [29, 30]. The biological significance of this effect is unknown, but a study comparing honey bee mouthpart microbiomes in propolis-rich and propolis-poor conditions suggests that propolis promotes the growth of putatively beneficial microbes, and may mitigate the growth of opportunistic microbes that trigger the production of antimicrobial peptides and other honey bee immune defenses [29]. If dysbiosis negatively impacts honey bee health, as studies of the honey bee gut have suggested [31], then the stabilization of microbial communities in propolis-rich environments could help explain why the presence of propolis supports bee resistance to external stressors.

Although abundant laboratory and colony-level evidence demonstrates that the propolis envelope supports honey bee health in a variety of ways, this natural tool for honey bee defense has yet to be integrated into commercial beekeeping operations. Borba et al. [18] made important strides in this direction, demonstrating that placing commercially produced plastic propolis traps on the interior walls of bee boxes stimulates bees to build a natural propolis envelope, which leads to measurable improvements in colony health. Unfortunately, plastic propolis traps are bulky. When attached to the inner walls of a beehive, they take up space and make it difficult to maneuver frames. They can also be expensive to implement on a large scale (US $11.50/propolis trap (Mann Lake Ltd, MN, USA, part no. HD370) x four traps/colony to cover the inner walls of just one brood chamber = US$46.00/colony), and this may represent a significant barrier for commercial beekeepers. In recent years, surface texture treatments like rough wood, saw kerfs, screen walls, and grooved aluminum plates have been tested by both bee researchers and beekeepers [32–34]. These textures do stimulate propolis deposition in beehives [34] and could represent a viable alternative to propolis traps. However, their impacts on colony health have not yet been tested, nor has the effect of increased propolis deposition been examined in a real-world commercial beekeeping setting.

Our study addressed two main questions: (1) How do rough wood boxes compare to boxes outfitted with propolis traps in terms of their ability to stimulate propolis collection and support colony health? And (2) can rough wood boxes support colony health in both stationary and migratory commercial beekeeping contexts? To answer these questions, we conducted two experiments. To address question one, we compared propolis deposition and colony health in rough wood boxes, boxes outfitted with propolis traps (proven to support colony health by Borba et al. [18]), and smooth wood control boxes in a stationary apiary setting. To address question two, we collaborated with a large commercial beekeeping operation to evaluate propolis deposition and colony health in rough wood and control boxes in a migratory beekeeping context over multiple years. Lastly, we conducted landscape analyses to shed some light on potential differences in the diversity and abundance of resin resources in the areas surrounding stationary and migratory beekeeping yards.

## Materials and methods

### Colony set-up

We evaluated propolis deposition and colony health across multiple hive types in stationary (2019–2020) and migratory (2019–2020, 2020–2021) beekeeping contexts in two complementary, but distinct experimental setups. In our stationary yard, we compared three texture treatment types: 1) plastic propolis traps (Mann Lake Ltd, MN, USA, part no. HD370) stapled to the four interior walls of each standard Langstroth-size deep hive body, following Borba et al.

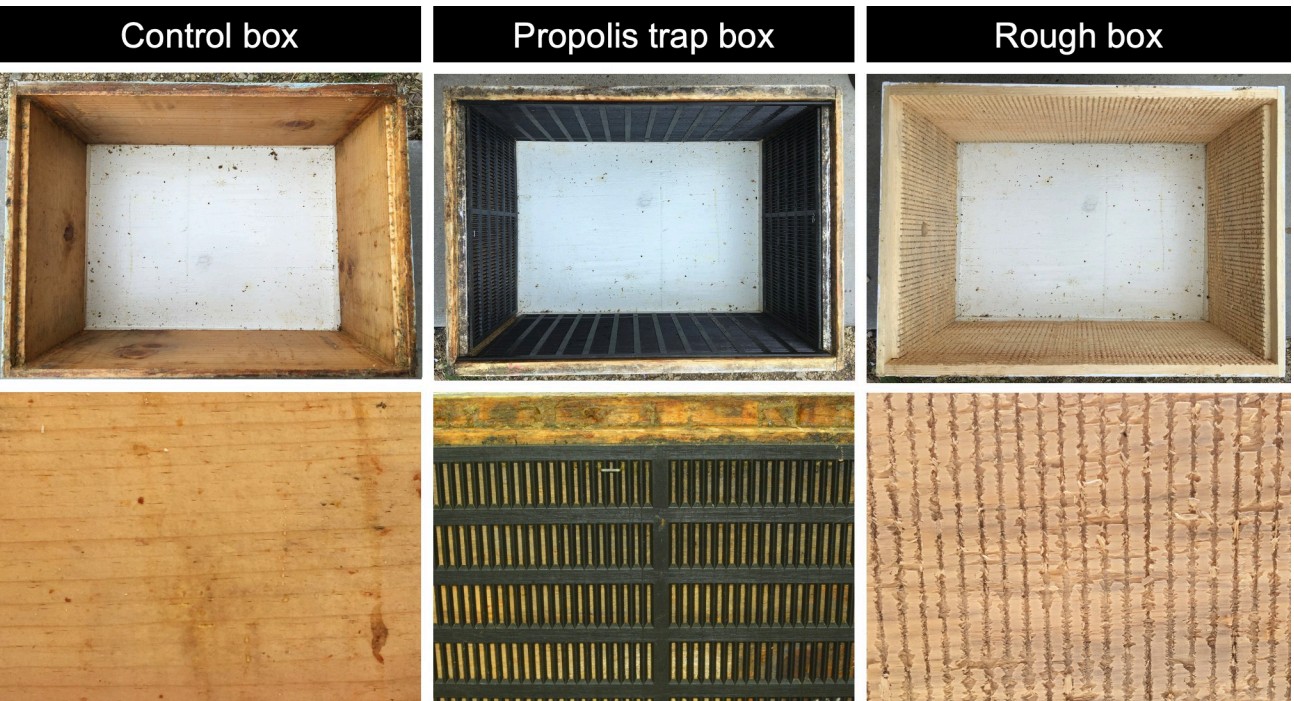

**Fig 1. Interior hive surface textures modified to stimulate propolis collection.** Three box types were evaluated: unmodified, smooth wood boxes ("control boxes"), smooth wood boxes outfitted with plastic propolis traps ("propolis trap boxes"), and boxes with rough, grooved interior walls ("rough boxes"). Control boxes consisted of previously used standard bee boxes scraped clean prior to installation to remove all visible traces of propolis. Propolis trap boxes contained propolis traps cut to hive body dimensions, stapled to all four interior hive walls, following Borba et al. [18]. Rough boxes (Propolis Hive Company, MN, USA) contained deep vertical grooves measuring 0.3175 cm wide by 0.3175 cm. Grooves were cut every 0.635 cm into the interior surface of all four walls of the deep hive body. The inner surface of these grooved hive bodies was not planed or sanded so it was rough, even slightly splintery.

[18] (propolis trap boxes), 2) roughened wood Langstroth-size deep hive boxes (Propolis Hive Company, MN, USA) specially constructed to provide bees with a highly texturized interior surface (rough boxes), and 3) standard, smooth Langstroth-size deep hive boxes scraped clean prior to installation to remove all visible traces of propolis (control boxes) (Fig 1). Rough boxes contained 0.3175 cm wide by 0.3175 cm deep grooves cut vertically and spaced every 0.635 cm on the interior surface of all four walls of the hive body. These grooves were designed to mimic the grooves found inside hollow tree cavities. The inner surface of these grooved hive bodies was not planed or sanded and remained rough.

The proof-of-concept experiment we conducted in our stationary yard allowed us to determine whether rough boxes were as effective as previously tested propolis trap boxes in stimulating propolis collection and supporting colony health [18]. Separately, we evaluated the rough box design in a migratory beekeeping operation, monitoring propolis deposition and colony health over the course of two years. Although similar methodologies were applied in both stationary and migratory contexts, the central aim of this study was to compare propolis deposition and colony health across box types, rather than across contexts (e.g., stationary/migratory).

**Stationary colonies.** The stationary component of this study was conducted in 2019–2020 at Carver Park Reserve, MN, USA (44.885776, -93.703419). Packages (Olivarez Honey Bees, Inc, California, USA) (n = 32) containing Saskatraz queens were introduced in April of 2019 in 10-frame Langstroth hive boxes featuring the three surface texture treatments described in Fig 1 (10 rough box hives, 10 propolis trap hives, and 12 control hives).

**Migratory colonies.** The migratory component of this experiment was conducted over the course of two years from 2019–2021 in collaboration with a large-scale commercial bee-keeper. Queenless colony divisions were created in 10-frame deep Langstroth hive boxes with smooth hive walls with two frames of sealed brood, two empty combs, four combs of honey and pollen, and a plastic frame feeder in southern Mississippi in March 2019. Queens were grafted from Italian breeder queens selected from within the operation. Queen cells were installed into queenless colony divisions one day before emergence. Newly emerged queens were allowed to open mate in an exclusive drone saturation area established by the beekeeper. Colonies were inspected in late April 2019 to ensure mating success as identified as colonies that had areas of sealed brood consistent with time from emergence and mating. Queens were marked on the notum with an approximately 3mm dot from a pink Sharpie® oil-based paint pen for later identification and assessment of queen replacement events. Colonies were inspected for amount of sealed brood and adult bee population (frames of bees) [35]. A total of 120 colonies were included in year one of this study (2019–2020); frames were moved between colonies to ensure similar amounts of sealed brood and adult bee population. The control colonies (n = 60) were housed in the existing hive bodies with smooth interior walls. Experimental colonies (n = 60) were transferred to rough box hive bodies (rough boxes) as described above. Propolis trap colonies were not included in the migratory component of this study due to the cost of installing propolis traps at scale, and the difficulties these present for beekeepers manipulating frames. Control and rough box colonies were housed in 10-frame deep Langstroth boxes in which the bottom two boxes contained eight frames of drawn comb with a 5 cm wide deep frame feeder. Colonies in the second year of this study (2020–2021) were set up and standardized following the same practices, but housed in rough boxes derived from the first year of the study which already contained some amount of propolis.

## Colony management

**Stationary colonies.** Colonies were given routine management on a bi-weekly basis during the growing season from spring 2019 to spring 2020. In April, new package bees were fed a patty of Ultra Bee High Protein pollen substitute (Mann Lake Ltd, MN, USA, part no. FD374) and 50% w/v sugar syrup via top feeder. As colonies grew, a second hive body (brood box) with the corresponding surface texture treatment was added. In June and July, medium supers (no texture treatment) were added for honey storage as needed. Colonies were inspected regularly for disease and treated for *Varroa* mites in late August/early September with Formic Pro® (NOD Apiary Products, Ontario, Canada) according to the labeled instructions: one strip was applied between brood chambers for ten days and then replaced with another strip for another ten days after which the strip was removed. Colonies were fed 50% w/v sugar syrup in the fall and wrapped with Bee Cozy Winter Wraps (Mann Lake Ltd, MN, USA, part no. WT160) in October. Colonies that survived winter were noted in spring of 2020.

**Migratory colonies.** All management of colonies followed the cooperating beekeeper's proprietary standard practices as described below. Colonies were initiated in southern Mississippi in March 2019 and maintained until they were transported to South Dakota in early May 2019. In South Dakota, colonies were distributed among four different apiaries. Each apiary contained 15 control and 15 experimental colonies as well as 34 colonies unrelated to the study for a total of 64 colonies per apiary. Additional rough boxes were provided to the beekeeper to add to the colonies after transport from Mississippi to South Dakota so the rough box treatment surrounded the brood chamber. Honey supers placed above the two-box brood chamber in both the control and rough box colonies were 10-frame deep Langstroth boxes with smooth walls. Boxes in both the control and experimental groups were added and

removed at the discretion of the beekeeper throughout the season. Following the beekeeper's management strategy, colonies in the study were provided 50% w/v sugar syrup in the feeders and a 500g soya-based protein supplement patty manufactured in-house according to the beekeeper's proprietary formula upon arrival to South Dakota in May 2019. Supplemental feeding of syrup and protein supplements were provided at the beekeeper's discretion throughout the season. Honey was harvested in mid-August 2019. Immediately after honey harvest, all colonies were condensed to two boxes, provided syrup and a supplemental protein patty as described above. Colonies were inspected and sampled immediately after honey harvest and then treated for *Varroa* mites following proprietary practices. Colonies were transported from South Dakota to holding yards in California in late October 2019 where they were provided syrup and supplemental protein patties at the beekeeper's discretion as described above. Colonies housed in two deep boxes were moved into almond orchards in early February 2020 and inspected in mid-February 2020. Colonies were returned to Mississippi in mid-March 2020 and inspected for the final time in late March 2020. Colonies were managed in the same manner in year two of this study, from 2020–2021.

## Landscape composition

To determine whether differences in propolis deposition between stationary and migratory colonies corresponded to differences in resource availability, we characterized the landscapes surrounding each of the apiary locations used during year one (one stationary yard, four migratory yards). Landscape data was pulled from the USDA-NASS Cropscape database's 2019 Cropland Data Layer (https://nassgeodata.gmu.edu/CropScape/). A circle with a 2.5-mile (4 km) radius was drawn around each apiary (corresponding to honey bees' typical foraging range), and land use statistics were calculated within these defined areas of interest. Land use types were sorted into the following categories: grass and pasture, forest and shrubs, water, herbaceous and woody wetlands, developed, corn and soy, and other crops. Proportional land use was calculated by dividing each category's acreage by the total acreage within the 2.5-mile (4 km) radius. Apiary locations are not disclosed here in order to protect the privacy of the beekeeper who participated in this study.

## Colony-level measurements

**Stationary colonies.** Colonies were assessed in August of 2019 and monitored for survival through the spring of 2020. In August, frames of bees were counted for both the top and bottom hive body. Queen status (i.e., whether the colony contained a living queen, and whether this queen was the same queen the colony had at the beginning of the experiment) was ascertained, and brood pattern was evaluated using a scale commonly used by beekeepers (1 = poor, 2 = fair, 3 = good). Brood frames were inspected for signs of brood disease (e.g., American foulbrood, European foulbrood, chalkbrood) and parasitic mite syndrome. Honey supers (i.e., the boxes located at the top of the hive where the bees store excess honey) were removed and weighed, and the weight of an average empty honey super (calculated by averaging the weight of ten empty supers) was subtracted from each to determine the approximate weight of the honey inside. Propolis deposition was measured using a visual scoring system (see "Propolis deposition scoring" section below). Only strong, healthy colonies (30/38 colonies) with greater than twelve frames of bees and no signs of parasitic mite syndrome or brood disease were scored for propolis deposition and used for immune gene expression analysis.

**Migratory colonies.** In year one, colonies were inspected in April of 2019 during colony establishment in southern Mississippi, in August of 2019 immediately after honey harvest in South Dakota, in February of 2020 during almond pollination in California, and in March of

2020 in southern Mississippi after almond pollination. Adult bee population (frames of bees) and amount of sealed brood were visually estimated using standard methods [35], and total bee population was calculated by adding frames of bees and frames of brood. The status of the queen bee was determined by the presence or absence of a paint-marked queen. *Varroa* infestation was measured by collecting approximately 300 bees from frames of sealed brood into a 1L ziptop bag and transporting them back to the USDA-Honey Bee Lab in Baton Rouge where *Varroa* were dislodged by shaking the bees in soapy water on an oscillating table shaker for >30 minutes. The number of *Varroa* and honey bees in the sample were counted and *Varroa* infestation was calculated as the number of *Varroa* mites per 100 bees. Honey production (kilograms/colony) was measured by weighing each honey super containing and subtracting the weight of an empty honey super. Propolis deposition was measured using a visual scoring system (see "Propolis deposition scoring" section below). Colony survivorship was measured in February of 2020, when migratory colonies were transported to California for almond pollination. Survivorship was calculated as the number of colonies remaining in the study in February relative to the starting number of colonies (n = 120 overall, n = 60 in each of the two treatment groups). Measuring survivorship in migratory operations can be complicated, since beekeepers regularly combine or otherwise alter weak colonies. Thus, the discontinuity of a colony could signal either a colony death or a management intervention. Brood was inspected qualitatively and disease and brood issues were noted. Signs of European foulbrood (EFB) (*Melissococcus plutonius* ex White 1912) infections were noted in March of 2020 and EFB infection scores were calculated per brood frame following established protocols (0 = no cells, 1 = less than 10 cells, 2 = 11–100 cells, 3 = more than 100 cells).

Travel and work restrictions due to the COVID-19 pandemic limited the scope of work performed in year two of the migratory study. Colony establishment was performed in March and April 2020 as described above. The initial inspection in Mississippi in April 2020 only included data on frames of bees and frames of brood. Honey production was measured in South Dakota in September 2020. Queen status, frames of brood, frames of bees, brood pattern (ranked 1–5 with 1 being poor and 5 being a solid brood pattern), were measured during almond pollination in California in February of 2021 by contracted members of the Bee Informed Partnership's Tech Transfer Team. Brood amount was scored on a 5-point scale compared to the amount of brood area following their standard protocol. Propolis deposition score was measured for surviving colonies that were transported back to Mississippi in March of 2021.

## Propolis deposition scoring

In year one, for both stationary and migratory colonies, propolis deposition was assessed within one week of collecting bee samples for gene expression analysis. Frames were removed from hive bodies, and all four walls of the second deep were photographed (Fig 2). For propolis trap colonies, traps were removed from the hive walls, and photographs were taken both of the wall and of the detached trap, in order to account for all propolis deposited. Propolis deposition was evaluated following Hodges et al. [34]. Four observers scored each photo on a scale from one to ten, where one is 0–10% wall coverage and ten is 90–100% wall coverage (S1 Fig), and an average score was obtained for each box (S2 Fig). A Fleiss kappa test indicated a fair amount of agreement between evaluators (Kappa = 0.25).

**Stationary colonies.** Propolis deposition was scored in August of 2019, after colonies had been established for four months.

**Migratory colonies.** Propolis deposition was scored at three time points. Year one colonies were scored in August of 2019 and February of 2020. A subset of year two colonies (which were established in the same control and rough boxes used for year one bees) were scored in

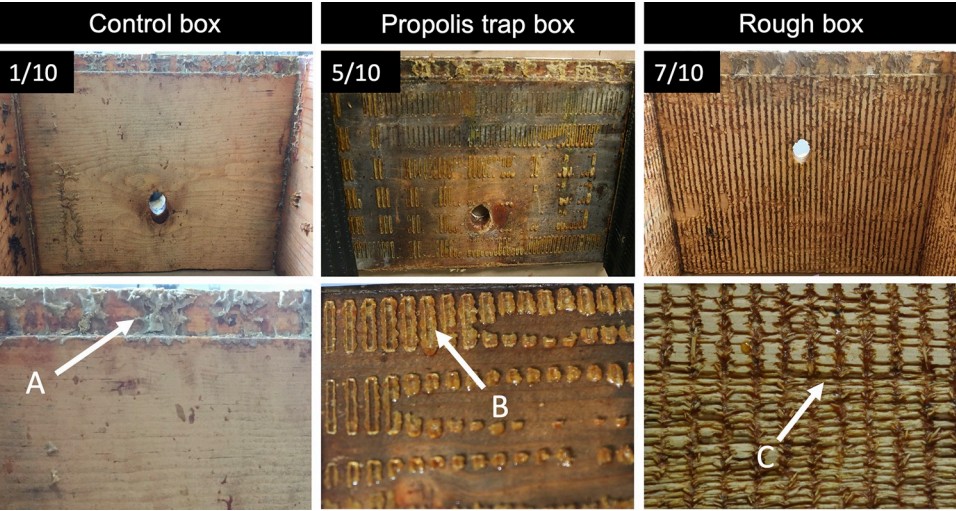

**Fig 2. Propolis deposition scoring.** The interior walls of control, propolis trap, and rough box colonies were photographed, and propolis deposition was scored on a scale from 1–10, where one is 0–10% wall coverage and ten is 90–100% coverage. In control colonies, propolis was primarily deposited on frame rests (A), the ledges that support the frames that bees build combs on. In propolis trap colonies, propolis was deposited in the small, rectangular holes in the propolis traps; these rectangular deposits (B) generally remained fixed to the hive walls even after propolis traps were removed. In rough box colonies, propolis was deposited in the cracks and crevices that covered the hive walls (C).

March of 2021 after they returned to Mississippi from almond pollination. COVID-19 restrictions during 2020–2021 made it impossible to monitor propolis deposition on a more regular basis in migratory colonies.

## Sample collection for gene expression analysis

**Stationary colonies.** Newly emerged bees (aged approximately one day) were paint-marked on the thorax using enamel paint (Mann Lake Ltd, MN, USA, part no. Z349) and recovered from the colony six days later. Twenty 7-day-old bees were collected per colony, stored in 15mL Falcon tubes on dry ice, and then transferred to a -80˚C freezer until processing. Seven-day-old bees were used because immune expression is less variable in young bees; variation in immune gene expression increases when bees leave the hive to forage [36, 37].

**Migratory colonies.** Due to the large number of migratory colonies involved in this experiment, paint-marking and collecting 7-day-old bees from migratory colonies was not practical. Therefore, at each sampling interval (August 2019, February 2020, February 2021), a sample of bees from frames of sealed brood were collected into a 1L ziptop bag and placed immediately on dry ice in a cooler while in the field. Samples were transported on dry ice back to the USDA Honey Bee lab in Baton Rouge, LA where they were stored at -80˚C until molecular analyses could be conducted with a subsample of 30 bees used for gene expression.

## Real-time PCR methods

**Stationary colonies.** RNA was extracted from individual whole bee samples (20 bees/colony) at the University of Minnesota Bee Research Facility. Bees were homogenized in microcentrifuge tubes using a pestle. RNA was extracted using the reagent TRIzol (Ambion, Austin, TX, USA), following the protocol recommended by the manufacturer [38]. A NanoDrop2000 (Thermo Scientific Inc., Grand Island, NY, USA) was used to determine the quality and quantity of the RNA extracted. DEPC treated water was added to samples to normalize RNA

concentration at 100ng/μl. Samples were stored at -80°C and shipped to USDA facility in Baton Rouge, Louisiana for cDNA synthesis and qPCR to quantify the expression of immune genes *abaecin*, *AmEater*,*AmPPO*, *defensin-1*, *hymenoptaecin* and *relish*, as well as reference genes *pros54* and *ß-actin* (S1 Table). cDNA synthesis was completed using QuantiTect Reverse Transcription Kits (Qiagen) with 2 μg of RNA, following the manufacturer's protocol. qPCR was performed on 1-μl aliquots of each sample, in triplicate, in a total reaction volume of 10 μl, utilizing SsoAdvanced Universal SYBR Green Supermix (Bio-Rad) on Multiplate 96-well optical PCR plates (Bio-Rad). All analyses were run on CFX Connect Real-Time PCR Detection Systems (Bio-Rad), using previously optimized thermal protocols (S1 Table).

*Migratory colonies and qPCR*. Pools of 30 whole bees were randomly subsampled from the 300 bees collected as described above, placed into 30 mL tubes (19-6358Z, Omni), and homogenized using a Bead Ruptor Elite (Omni). RNA was extracted using the Maxwell RSC SimplyRNA extraction kit (Promega) following the manufacturer's protocol. RNA quality and quantity was assessed using a NanoDrop One. cDNA synthesis and qPCR were conducted as described above. qPCR was used to quantify the expression of immune genes *defensin-1*, *abaecin*, *hymenoptaecin*, *AmPPO*, and *AmEater*; bacteria *Bartonella apis*, *Bifidobacterium asteroides*, *Lactobacillus* Firm-4 phylotype, *Lactobacillus* Firm-5 phylotype, *Snodgrassella alvi*, and *UniBact*, a primer coding for a universal bacterial gene sequence; viruses acute bee paralysis virus (ABPV), black queen cell virus (BQCV), chronic bee paralysis virus (CBPV), deformed wing virus A (DWV-A), deformed wing virus B (DWV-B), Israeli acute paralysis virus (IAPV), Kashmir bee virus (KBV), Lake Sinai virus 1 (LSV-1), and Lake Sinai virus 2 (LSV-2); genes associated with European foulbrood; as well as reference genes p*ros54* and *ß-actin* (S1 Table).

## Statistical analysis

All statistical analyses were performed using R Statistical Software (v4.2.1; R Core Team 2022).

**Landscape analysis.** We calculated percent cover of herbaceous and woody wetlands and forest and shrubs in the areas surrounding the stationary and migratory yards. Unlike water, grass and pasture, corn and soy, and other crops, these landscape types are likely to contain resin-producing plants [39, 40]. Because the presence of resin resources in developed land varies depending on the type of development, this landscape type was excluded from analysis. We used a simple linear model to determine the correlation between the presence of landscapes likely rich in resin resources (percent cover) and propolis deposition score.

**Colony-level measures.** We assessed the effect of box type on multiple colony-level response variables, including propolis deposition, number of frames of adult bees, total bee population (i.e., number of frames of bees + number of frames of brood), honey production, *Varroa* load, brood disease, and survival after one year. For stationary colonies, we used ANOVA to determine the effect of box type (i.e., rough, propolis trap, control) on propolis deposition, frames of bees, honey production, and survivorship. We then used post-hoc two-tailed t-tests with a Bonferroni adjustment to determine differences between treatments.

For migratory colonies, we generated a mixed-effects model to compare propolis deposition across box type, and across multiple time points. Fixed effects included box type, sample date, and the interaction between sample date and box type, and random effects included colony. We used ANOVA to determine the effect of box type on total bee population, *Varroa* load, signs of EFB, colony survivorship, overall honey production, and honey production by colony size. Finally, we ran a correlation analysis to determine whether varroa levels were correlated with virus levels in migratory colonies.

**Gene expression.** Ct values were determined using the Bio-Rad CFX Maestro™. ΔCt was calculated for each target gene by subtracting the average Ct for reference genes *Pros54* and *ß-*

*actin* from the target gene Ct. For stationary colonies, samples with reference gene Ct values greater than 30 or less than 23.5 were excluded from analysis. For migratory colonies, samples with reference gene Ct values less than 25 were excluded from analysis. Seven out of 225 data points were excluded from the stationary data set and six out of 243 data points were excluded from the migratory data set.

$$\Delta Ct = target\ gene\ Ct - \bar{x}(reference\ genes)\ Ct$$

Gene expression was calculated using the transformation $2^{-\Delta Ct}$ following Schmittgen and Livak [41]. Since this transformation results in a non-normal distribution of linear model residuals, data were log-transformed ($log(2^{-\Delta Ct})$) for all statistical analyses.

We fit Bayesian linear mixed-effects models to describe the relationship between propolis score and both the mean and standard deviation of gene expression for honey bee immune genes, bacterial genes, and viral genes for all colonies. In these distributional models, both the mean and standard deviation of gene expression were allowed to vary with propolis score. We fit random intercepts to each colony and to each date nested within yard. To make inferences about whether differences in propolis score were associated with differences in mean gene expression, we examined the posterior distribution of the slope parameter of the mean. Similarly, to make inference about whether differences in propolis score were associated with differences in the variability of gene expression, we examined the posterior distribution of the slope parameter of the standard deviation. A negative slope parameter for the standard deviation indicates that as propolis score increases, variability in gene expression decreases; this can be interpreted as a stabilizing effect of propolis on gene expression. To obtain point estimates of these parameters we used the median of the posterior distributions, and to assess uncertainty in our estimates we computed 66%, 90%, and 95% quantile credible intervals of the posteriors.

This analysis was done using Stan software version 2.30 (Stan Development Team [42]) and the R packages cmdstanr [43], brms [44], and bayestestR [45].

## Results

### Landscape analysis

The landscape surrounding the stationary yard consisted of forest and shrubs (25%), herbaceous and woody wetlands (18%), water (18%) and developed land (13%), with crops representing only 12% of the total landscape. Landscapes surrounding year one migratory yards were dominated by grass and pasture (percent land use > 50%) (Fig 3). Crops covered 19–38% of these landscapes, with corn and soy plantings representing 42–71% of total crop cover. Our simple linear model indicated that across both stationary and migratory landscapes, propolis score was positively correlated ($r = 0.94$, $p = 0.02$) with percent cover of herbaceous and woody wetlands and forest and shrubs, landscapes likely rich in resin resources.

### Propolis deposition

Bees deposited more propolis in rough boxes than in other box types (Fig 4). In stationary colonies, just four months into colony development, propolis score averaged 7.5 (SE = 0.2) in rough boxes. This score was significantly higher than the 4.9 (SE = 0.2) average in propolis trap boxes, ($t_{27} = 10.1$, $p < 0.001$) and the 1.7 (SE = 0. 1) average in control boxes ($t_{27} = 21.9$, $p < 0.001$). Propolis score was also significantly greater in propolis trap boxes compared to control boxes ($t_{27} = -11.8$; $p < 0.001$).

In migratory colonies, our mixed-effects model indicated that propolis deposition was significantly affected by both box type ($t_{167} = -9.1$; $p < 0.0001$) and the interaction between

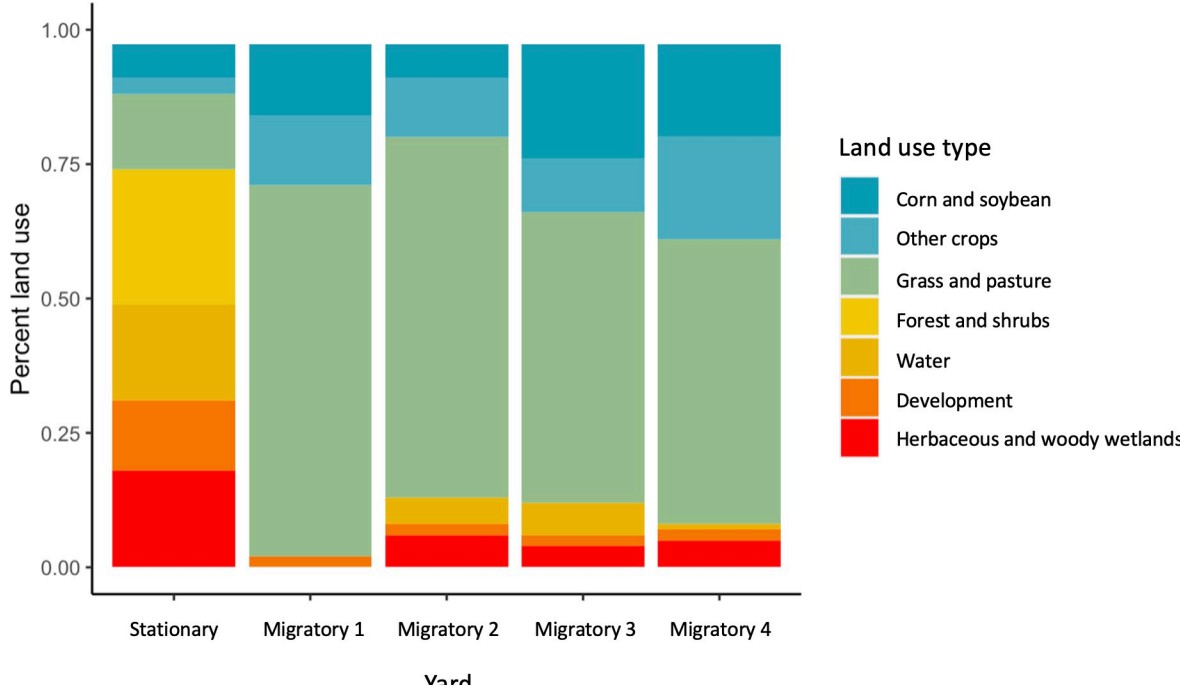

**Fig 3. Land use in landscapes surrounding stationary and migratory bee yards.** Land use for areas surrounding bee yards (radius = 2.5 mile, 4 km) was analyzed using the USDA-NASS Cropscape database's 2019 Cropland Data Layer. Landscapes surrounding migratory yards were dominated by grass and pasture. Landscape surrounding the stationary yard contained higher percentages of forest and shrubs, herbaceous and woody wetlands, water, and development. Propolis score was positively correlated ($r = 0.94$, $p = 0.02$) with percent cover of herbaceous and woody wetlands and forest and shrubs, landscapes likely rich in resin resources.

box type and sample date ($t_{167} = 9.3$; $p < 0.0001$). Bees deposited more propolis in rough boxes compared to controls for all dates, and this difference grew more pronounced over time. In August of 2019, propolis score averaged 3.2 (SE = 0.2) in rough box colonies, which was significantly higher than the 2.0 (SE = 0.2) average in control colonies ($t_{149} = 5.5$; $p < 0.0001$). By February of 2021, propolis scores had more than doubled to an average of 7.2 (SE = 0.4) in rough box colonies ($t_{138.8} = 11.7$; $p < 0.0001$) but remained stagnant at 2.2 (SE = 0.3) in control colonies ($t_{142} = 142$; $p = 1$).

Propolis deposition was higher in rough box stationary colonies (propolis score 7.5; SE = 0.2) compared to rough box migratory colonies (propolis score 3.2; SE = 0.2) in August of 2019 when all colonies were evaluated ($F_{1,102} = 72.5$, $p < 0.0001$). Migratory rough box colonies took over a year to achieve the levels of propolis deposition that stationary colonies achieved in just four months.

## Colony size

For both migratory and stationary colonies, frames of bees were counted for all colonies at all sampling dates; for migratory colonies frames of brood were counted at multiple time points during both years of the experiment. Where possible (August of 2019, February of 2020, February of 2021), we combined frames of brood and frames of bees to calculate "total bee population".

There were no significant differences in the number of frames of bees across treatment for the stationary colonies (Fig 5; $F_{2,32} = 0.07$, $p = 0.9$). In the migratory operation, there were no differences across treatment in total bee population in August of 2019, but by February of

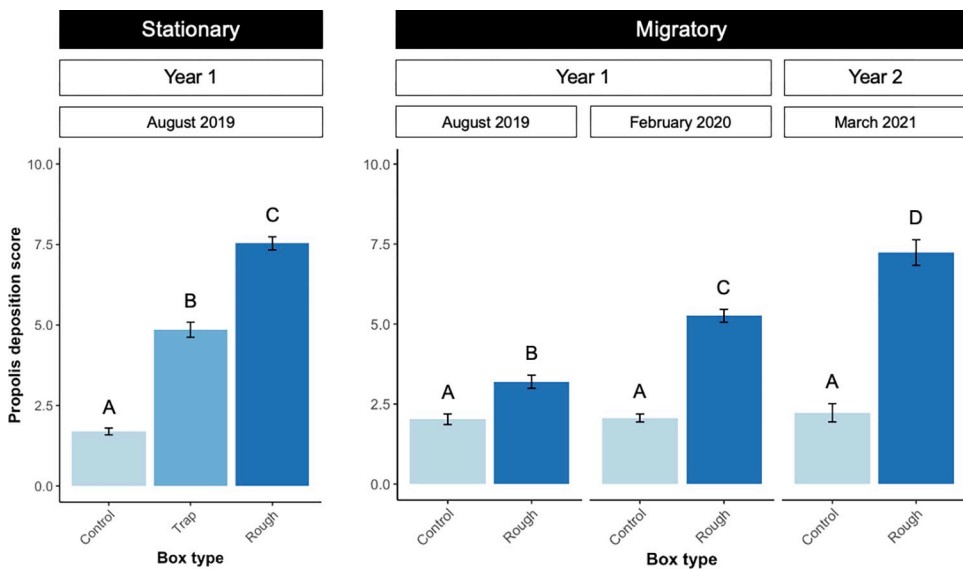

**Fig 4. Propolis deposition across box types in stationary and migratory contexts.** Propolis deposition on each interior brood chamber wall was scored on a scale from 1–10 where one is 0–10% wall coverage and 10 is 90–100% wall coverage. Scores were averaged to calculate each colony's "propolis score." Stationary colonies (n = 30) were evaluated in August of 2019, after four months of propolis deposition. Migratory colonies were evaluated in August of 2019 (n = 106), February of 2020 (n = 75), and March of 2021 (n = 27). Propolis score was higher in rough box colonies than in trap colonies and control colonies. Propolis score increased over time in rough box migratory colonies. Mean propolis score ± standard error is shown for each treatment. Letters indicate significant differences between treatments, and, in the case of migratory colonies, differences between years ($p < 0.05$).

2020, the total bee population in rough box colonies was significantly larger than in control colonies, by a margin of nearly two frames of bees plus brood ($F_{1,74}$ = 4.4, $p$ = 0.04). There were no statistically significant differences in total bee population across treatment in year two, though by the end of year two (February 2021), we observed a non-significant increase in total bee population in rough box colonies compared to control colonies ($F_{1,53}$ = 1.1, $p$ = 0.3).

## Immune gene expression

Propolis deposition had a seasonal effect on both the amount and variability of immune gene expression for multiple immune genes in stationary and migratory contexts (Fig 6; S2 Table).

For the August 2019 sample date in both stationary (n = 30) and migratory (n = 102) colonies, immune gene expression tended to decrease with increasing propolis score. In stationary colonies, our distribution model provided strong evidence for a negative correlation between propolis score and *defensin-1* expression. This model also provided some evidence that *relish*, *hymenoptaecin*, and *AmEater* expression decreased with increasing propolis score. However, propolis score was positively correlated with *AmPPO* expression.

In migratory colonies, our distribution model provided moderate evidence that *defensin-1* expression tended to decrease with increasing propolis score, and some evidence that *abaecin* expression tended to decrease with increasing propolis score.

Our distribution model also provided some evidence that immune gene expression stabilized as propolis score increased. Variation in *hymenoptaecin*, *AmEater*, and *abaecin* expression tended to decrease with increasing propolis score in stationary colonies, as did variation in *abaecin* expression in migratory colonies. In contrast, in migratory colonies, our distribution model suggested that *AmEater* expression tended to destabilize with increasing propolis.

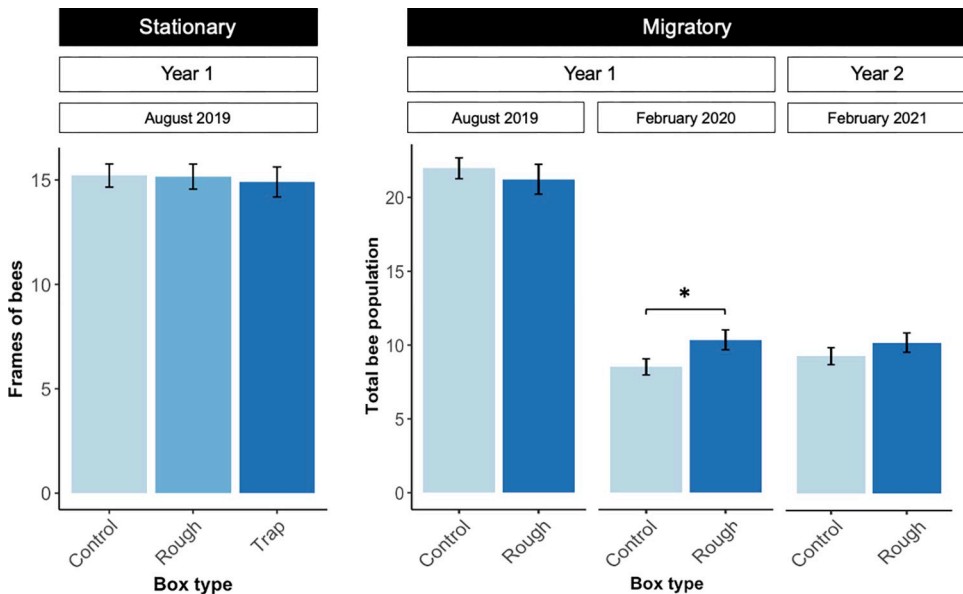

**Fig 5. Frames of bees and total bee population across box type in stationary and migratory contexts.** Frames of bees were quantified by counting the number of frames covered in bees in the first and second brood chambers. Total bee population was calculated by adding the number of frames of bees and the number of frames with brood present in the first and second brood chamber. Stationary colonies (n = 30) were evaluated in August of 2019. Migratory colonies were evaluated in August of 2019 (n = 110), February of 2020 (n = 76), and February of 2021 (n = 55). There was no difference between treatments in the number of frames of bees (stationary) or total bee population (migratory) in August of 2019. Total bee population was significantly higher in rough box colonies (mean number of frames = 10.4, SE = 0.7) compared to control colonies (mean number of frames = 8.5, SE = 0.5) at the end of year one ($F_{1,74}$ = 4.4, $p$ = 0.04). Mean number of frames of bees/total bee population ± standard error are shown for each treatment. Frames of brood are added to frames of bees for migratory colonies. Asterisks indicate significant differences between treatments ($p < 0.05$).

When we compared gene expression in individual bees from stationary colonies, we found convincing evidence that variation in *relish* expression decreased with increasing propolis score at the colony level, and some evidence that variation in *defensin-1* expression decreased with increasing propolis score at the colony level (Fig 7).

In migratory colonies, where bees were sampled for immune gene expression in August 2019, February 2020, and February 2021, the relationship between propolis score and immune gene expression differed among sample dates for some genes (Fig 8, S2 Table). *Defensin-1* expression tended to decrease with increasing propolis score in August 2019, but tended to increase with increasing propolis score in February 2020 and February 2021. For other genes, gene expression patterns were consistent among dates. Expression of *AmPPO* tended to increase and stabilize with increasing propolis score in February of 2020 and tended to increase in February of 2021. *AmEater* expression tended to destabilize with increasing propolis score in August 2019 and tended to increase and destabilize in February 2021. *Abaecin* expression tended to decrease and stabilize with increasing propolis score in August 2019, and tended to decrease in February 2021. There was no effect of propolis score on the expression of *hymenoptaecin* at any sampling date.

## Bacterial gene expression

Our distribution models indicated that propolis deposition likely had a seasonal effect on bacterial gene expression in migratory colonies (n = 102; Fig 8, S3 Table). In August of 2019, *B*.

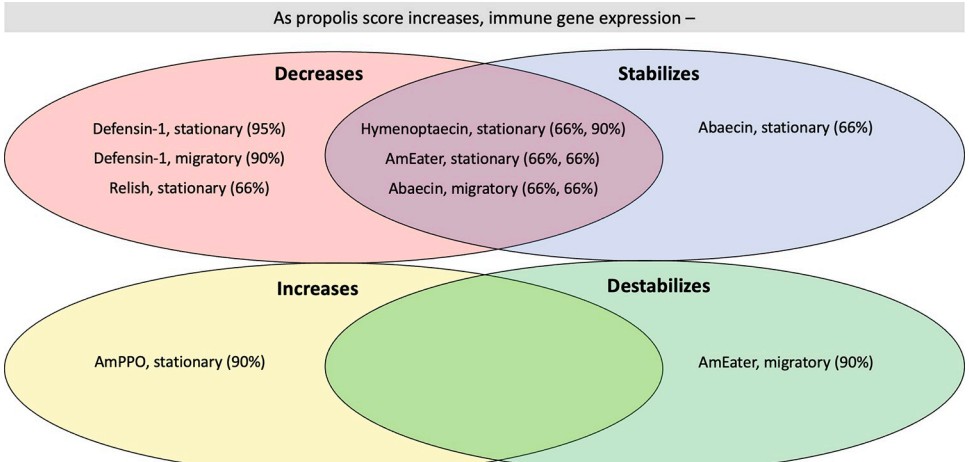

**Fig 6. Trends in immune gene expression with increasing propolis score for stationary and migratory operations, for August 2019 sampling date.** Gene expression in seven-day-old bees (stationary) and young bees collected from frames with sealed brood (migratory) was quantified using real-time PCR. Six immune genes (*abaecin*, *defensin-1*, *hymenoptaecin*, *relish*, *AmPPO*, and *AmEater*) were analyzed in stationary colonies (n = 30), and gene expression trends were analyzed at both the apiary and colony level. The same genes, with the exception of *relish*, were analyzed in migratory colonies (n = 102) at the apiary level. A distributional regression model was used to determine the probability that gene expression increases, decreases, stabilizes, or destabilizes with increasing propolis score (S2 Table). The percentages listed refer to the quantile credible interval, as determined by our model, and reflect the widest possible credible interval supporting the indicated trend (not containing zero). When two percentages are listed for one gene (e.g., *hymenoptaecin*, stationary (66%, 90%)), the first number listed corresponds to the grouping on the left (e.g., decreases); the second corresponds to the grouping on the right (e.g., stabilizes).

*asteroides*, *UniBact*, and Firm-5 phylotype expression tended to decrease with increasing propolis score. In February, *B. asteroides* and *UniBact* expression demonstrated the opposite tendency, increasing with increasing propolis score in 2020 and 2021, respectively. The effect of propolis deposition on *Bartonella* expression was fairly consistent across seasons; *Bartonella* expression tended to increase and stabilize with increasing propolis score in August 2019, and continued to increase with increasing propolis score in February 2020. Expression of the Firm-4 phylotype decreased with increasing propolis score in February 2021. There was no effect of propolis deposition on the expression of *S. alvi* at any time point in this study.

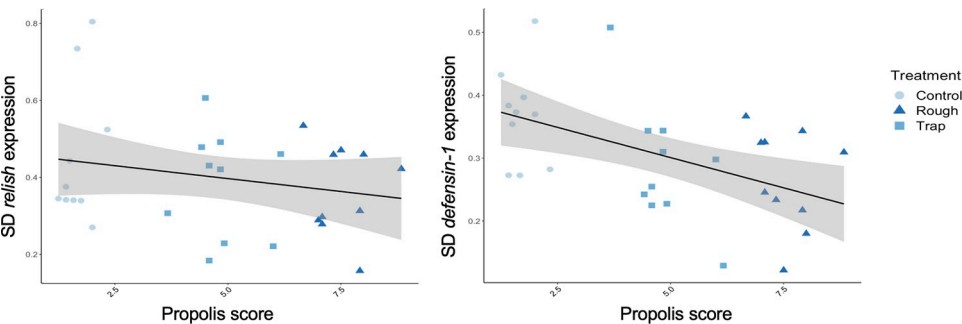

**Fig 7. Within-colony variation in immune gene expression across box types.** An average of seven bees per colony were processed to measure immune gene expression in stationary colonies using real-time PCR. Standard deviation in immune gene expression ($\log(2^{-\Delta Ct}$) was calculated for each colony, to determine whether variation in immune gene expression was correlated with propolis score. Standard deviation decreased with increasing propolis score for immune genes *relish* and *defensin-1*.

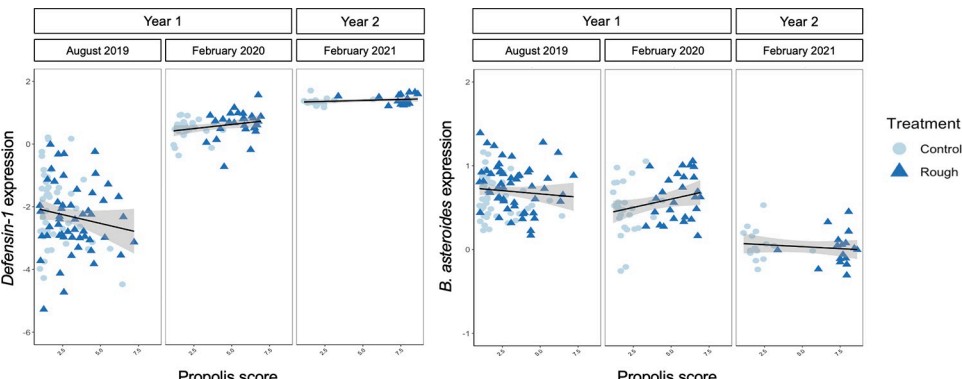

**Fig 8. Seasonal effects of propolis score on immune and relative bacterial gene expression in migratory colonies.** In some cases, genes whose expression tended to decrease and stabilize in August exhibited opposite trends in February. In migratory colonies, in August of 2019, expression of immune gene *defensin-1* (Median: -0.316, 90% QCI: [-0.6, -0.045]) and bacterial gene *B. asteroides* (Median: 0.065, 66% QCI: [0.017, 0.111]) tended to decrease with increasing propolis score, but in February of 2020 and 2021, *defensin-1* expression tended to increase with increasing propolis score (Median: 0.099, 90% QCI: [0.003, 0.199]; Median: 0.035, 66% QCI: [0.013, 0.053], respectively). *B. asteroides* expression tended to increase with increasing propolis score in February of 2020 (Median: 0.065, 66% QCI: [0.017, 0.111]).

## Pests and pathogens

The expression of *Melissococcus plutonius*, was significantly reduced in bees collected from rough box migratory colonies ($F_{1,77}$ = 5.66, $p$ = 0.02, Fig 9A). Although signs of European foulbrood (scores calculated based on number of symptomatic brood cells observed) were approximately 30% less severe in migratory rough box colonies than in migratory control colonies, these results were not significant ($F_{1,67}$ = 2.8, $p$ = 0.10, Fig 9B). Similarly, pre-treatment *Varroa* infestation (number of mites/100 bees) was lower in rough box colonies compared to control colonies by nearly one third, though this difference was non-significant ($F_{1,108}$ = 1.9, $p$ = 0.18, Fig 9C).

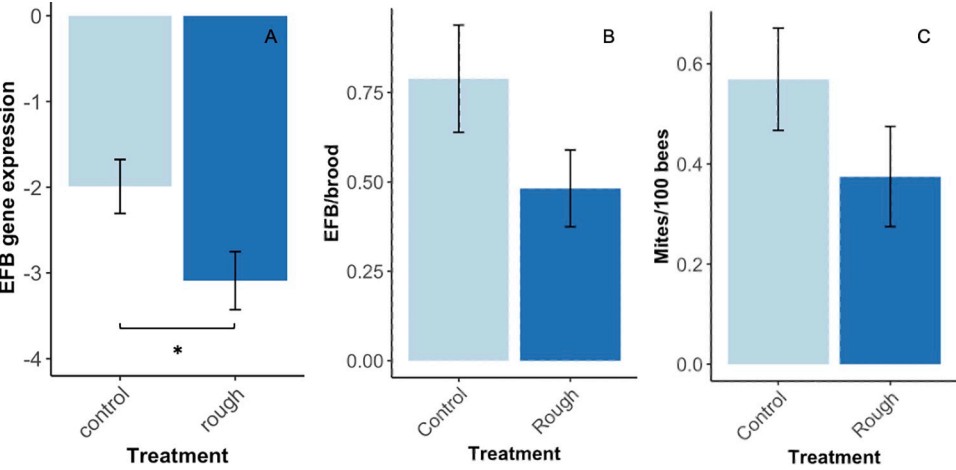

**Fig 9. Pathogen load in control and rough box migratory colonies.** (A) Detection of EFB via gene expression was significantly reduced in rough box colonies (n = 79, $F_{1,77}$ = 5.66, $p$ = 0.02) compared to control colonies in February 2020. Mean gene expression ($\log(2^{-\Delta Ct})$ ± standard error. (B) There was a marginal reduction in signs of EFB observed in March of 2020 at the colony level (n = 69, $p$ = 0.10). Mean EFB/brood ± standard error, where EFB score is divided by the number of frames with EFB present, and score is determined according to the following: 0 = no cells, 1 = less than 10 cells, 2 = 11–100 cells, 3 = more than 100 cells. (C) When colonies were sampled for *Varroa* mites in August of 2019 there were 35% fewer mites in rough box colonies than in control colonies, though this difference was not statistically significant (n = 110, $p$ = 0.18). Asterisks indicate significant differences between treatments ($p$ < 0.05).

Our distributional models provided some evidence that viral load for multiple viruses tended decrease with increasing propolis deposition (S4 Table). CBPV, IAPV, and LSV-1 decreased with increasing propolis deposition in August of 2019, and DWV decreased with increasing propolis deposition in both February of 2020 (DWV-A) and February of 2021 (DWV-A and DWV-B). In contrast, BQCV tended to increase with increasing propolis score in August of 2019. We detected no significant correlation between viral loads and mite levels when both were tested in August 2019 (CBPV: $r(105) = 0.05$, $p = 0.63$; IAPV: $r(105) = 0.13$, $p = 0.17$; LSV-1: $r(105) = 0.06$, $p = 0.53$; LSV-2: $r(105) = 0.04$, p = 0.65; BQCV: ($r(105) = 0.07$, $p = 0.49$);; ABPV: $r(105) = 0.08$, $p = 0.42$; DWV-A: $r(105) = 0.001$, $p = 0.98$; DWV-B: $r(105) = 0.13$, $p = 0.17$; KBV: $r(105) = -0.001$, $p = 0.99$.

### Survival

Only 13% of stationary colonies survived year one, with no differences in survivorship across treatments ($F_{2,34} = 0.37$, $p = 0.69$). High losses in the stationary yard were attributed to an issue with fall feeders, which prevented colonies from entering winter with sufficient food stores. In the migratory operation, only 58% of colonies (n = 231) survived and were deemed suitable to be sent to California for almond pollination services at the end of year one, but there were no differences in survival between rough box and control colonies ($F_{1,229} = 0.036$, $p = 0.85$).

### Honey production

In the stationary colonies, there were no differences in mean honey production across treatments (Fig 10; $F_{2,34} = 0.5$, $p = 0.6$). In the migratory colonies, rough box colonies (25 kilograms; SE = 3.9) produced 38% less honey than control colonies (39 kilograms; SE = 2.7) in year one ($t_{215.8} = 3.2$, $p = 0.02$); these differences corresponded to colony size. Large (>15 frames) rough box (45 kilograms; SE = 3.9) and control (48 kilograms; SE = 3.5) colonies were fairly even in terms of honey production, but small (<15 frames) control (34 kilograms; SE = 3.8) colonies produced, on average, 20 kilograms more honey than small rough box) 14 kilograms; SE = 2.9) colonies ($t_{102} = 4.5$, $p = 0.0001$, Fig 11). However, decreased honey production in small rough box colonies did not correspond to a significant increase in propolis deposition ($t_{102} = 2.0$, $p = 0.19$). In year two, when colonies were started in boxes used in year one that were already propolized and when honey production was higher overall, there was no effect of box type on honey production ($F_{1,102} = 0.075$, $p = 0.8$).

## Discussion

Our study evaluated strategies that beekeepers can use to support bees' construction of a natural, health-supportive propolis envelope. To date, propolis envelope support strategies have largely been tested in research settings and over relatively short periods of time. Here, we compared propolis deposition and colony health in rough wood boxes, boxes outfitted with propolis traps, and standard smooth wood boxes in a stationary context over one year and in a migratory beekeeping operation over two years. Our results provide convincing evidence that rough wood boxes are an effective means to stimulate propolis collection and support colony health and homeostasis in both stationary and migratory beekeeping contexts.

### Propolis deposition

Rough boxes were highly effective in stimulating propolis collection, compared to control boxes (stationary and migratory colonies) and boxes outfitted with propolis traps (stationary colonies). Stationary rough box colonies collected 50% more propolis than stationary colonies

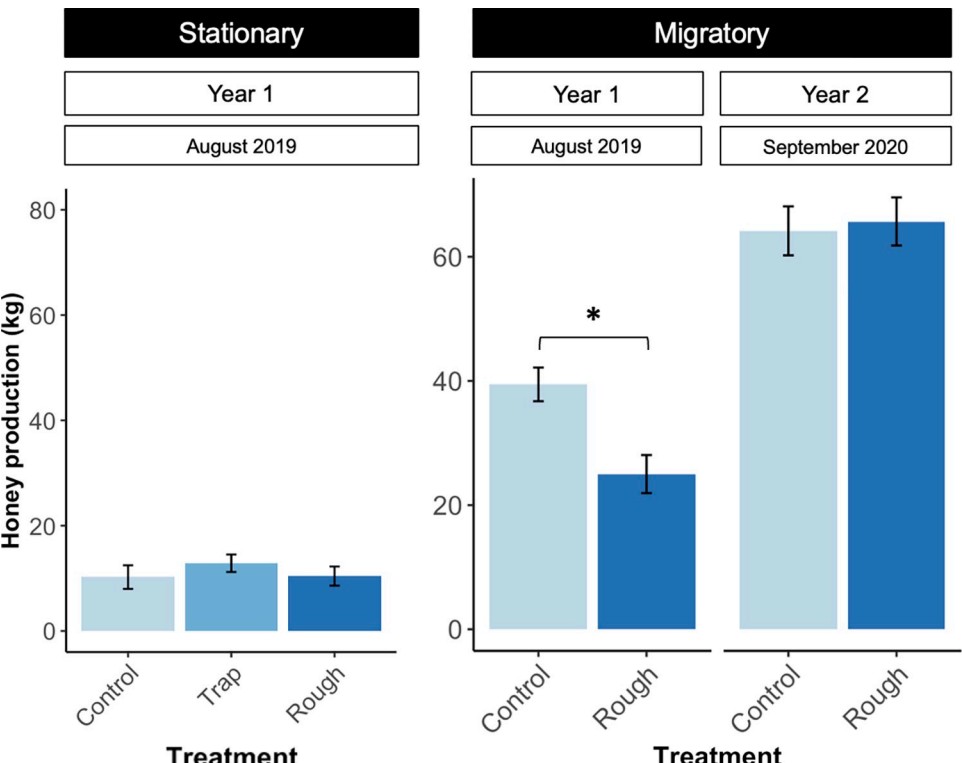

**Fig 10. Honey production across box type in stationary and migratory colonies.** Honey production did not differ between treatments in stationary colonies (n = 30). In migratory colonies, in year one (August of 2019), honey production was lower in rough box colonies by a margin of 15 kilograms (n = 112, $t_{215.8}$ = 3.2, $p$ = 0.02). By year two (September of 2020), there were no differences in honey production between treatments (n = 104). Mean kilograms of honey ± standard error is shown for each treatment. Asterisks indicate significant differences between treatments ($p < 0.05$).

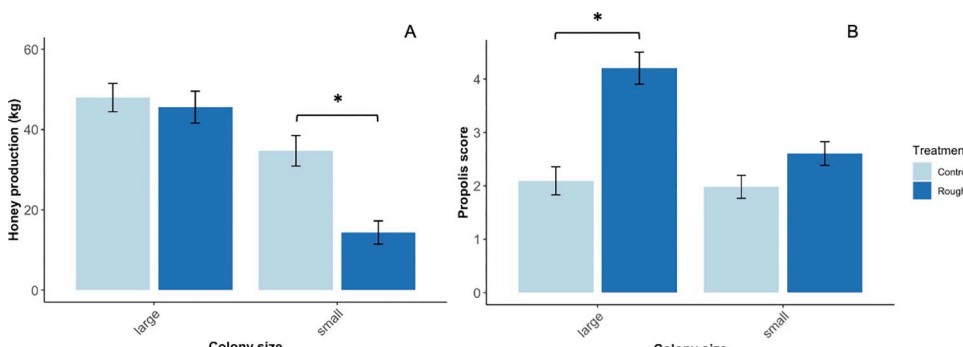

**Fig 11. Correspondence between colony size, honey production, and propolis collection in migratory rough box and control colonies.** Honey production (A) and propolis score (B) were quantified in August of 2019. Large rough box and control colonies (>15 frames) were fairly even in terms of honey production. Small control colonies (<15 frames), produced significantly more honey than small rough box colonies ($t_{102}$ = 4.5, $p$ = 0.0001). Large rough box colonies deposited significantly more propolis than large control colonies ($p < 0.0001$), but there was no difference in propolis deposition between small rough box colonies and small control colonies ($t_{102}$ = 2.0, $p$ = 0.19). Mean kilograms of honey and mean propolis deposition score ± standard error are shown for each treatment. Asterisks indicate significant differences between treatments ($p < 0.05$).

outfitted with propolis traps, demonstrating that rough boxes outperform this previously established method for supporting bee health [18, 46]. This result is in contrast with findings from Hodges et al. [34], where there were no differences in propolis deposition between rough box and propolis trap colonies. This discrepancy could be due to the fact that Hodges et al. [34] used boxes roughened with a mechanized wire brush, creating a two-dimensional rough surface. Our rough boxes contained texturized grooves, a three-dimensional rough surface which likely allowed for higher levels of propolis deposition. Future use of rough box colonies should strive to imitate the combination of rough wood textures, cracks, and crevices found in the hollow tree cavities where feral colonies nest.

In migratory colonies, bees deposited more propolis in rough boxes than in control boxes for all dates, and rough box propolis deposition increased over time, while control box propolis deposition remained stagnant. This suggests that, when provided with a stimulus, colonies continue to bring in resins to fully form and refresh the "propolis envelope." Notably, propolis build-up was slower in migratory rough box colonies than in stationary rough box colonies; it took migratory colonies nearly two years to come close to the amount of propolis that stationary colonies collected in just four months. Mountford-McAuley et al. [47] note that, in addition to box type, there are multiple factors that affect propolis production, among them resource availability and genetics. In our study, the landscape surrounding the stationary yard was more diverse than the landscape surrounding the migratory yards, with a notable presence of forest and shrubs and herbaceous and woody wetlands. The percent cover associated with these plant communities was significantly correlated with propolis deposition scores. Previous research has established that areas of high plant biodiversity tend to provide more resin resources than areas of low plant diversity, corresponding to increased propolis production [40]. Since different plant resins are effective against different pathogens, the implications of landscape composition could extend beyond propolis score [48]. Future studies should examine the ways in which landscape factors shape the composition of the propolis envelope (in addition to the amount of area it covers) and affect honey bee health.

Genetic differences may have also contributed to the variation we observed in propolis score, both between migratory and stationary yards, and between colonies in the same box type, in the same yard. Propensity for propolis collection is a highly heritable trait (coefficient of heritability = 0.87 [49]), and selection efforts can yield high-propolis colonies [50]. Different honey bee stocks were used in the stationary and migratory study, which, along with landscape differences, may have contributed to the variation in propolis deposition across contexts. Taken together, these findings suggest that, in order to fully realize the potential of the rough box, beekeepers must take steps to integrate resin resources into the landscape and, where possible, select for bees with propolis-collecting genetics. Still, even with unselected bees and across landscapes with varying levels of diversity, rough wood boxes supported improvements in multiple measures of colony health.

## Colony health

The use of rough boxes mitigates some types of pathogen pressure. In year one, rough *Varroa* mite loads were lower in migratory rough box colonies, compared to migratory control colonies. In our study, *Varroa* loads in migratory colonies were extremely low overall, with an average of approximately 0.5 mites per 100 bees. It is possible that these low numbers made it difficult to detect a significant contrast in mite infestation across treatments. Regardless, the marginal reduction in mite load that we did observe could be explained by recent findings from Pusceddu et al. [22], who found that the application of field-realistic quantities of propolis to artificial brood cells resulted in a near 20% increase in *Varroa* mortality during brood

rearing. In a previous study Pusceddu et al. [51] also observed an increase in resin foraging activity in *Varroa*-infested hives, pointing towards a possible example of social medication [52]. These effects have not been found to translate to reductions in mite loads at the colony level, possibly due to differences in the way researchers have attempted to simulate the propolis envelope. Previous studies have transplanted propolis harvested from one set of colonies to the tops of frames in a separate set of colonies to create a propolis-rich environment [20] or used propolis traps to encourage the formation of a natural propolis envelope [18], a strategy we now know to be less effective than the use of rough boxes. It is possible that a robust, honey bee-made propolis envelope helps mitigate *Varroa* load in ways that a human-made, or propolis trap-induced propolis envelope does not.

Although propolis has recently been shown to inhibit the growth of *M. plutonius in vitro* (Murray et al. BioRxiv [53]), to our knowledge, ours is the first study to observe lower levels of *M. plutonius* detection in propolis-rich (rough box) colonies, compared to propolis-poor (control) colonies. The significantly lower levels in *M. plutonius* gene expression that we observed in bees from rough box colonies compared to bees from control colonies corresponded to marginally lower levels of colony signs of EFB in rough box colonies compared to control colonies the following month. Future studies should investigate impacts of propolis-rich hive environments on this pathogen, taking into account both molecular methods and field observations in colonies experimentally challenged with EFB.

Propolis deposition also appeared to impact viral loads in migratory colonies. In August of 2019, viral loads for CBPV, IAPV and LSV-1 tended to decrease with increasing propolis deposition, and DWV tended to decrease in February of 2020 (DWV-A) and 2021 (DWV-A and DWV-B). Surprisingly, BQCV load tended to increase with increasing propolis deposition in August 2019. Previous studies have compared viral loads in bees from propolis-rich and propolis-poor environments and detected no differences [18, 21] or nuanced differences [20]. Notably, these studies compared viral loads across a propolis-rich/propolis-poor binary while our study examined viral loads along a propolis deposition gradient, which allowed us to more closely examine the relationship between propolis deposition and viral load. In addition, the quantile credible interval metric we used in our analysis is more expansive (i.e., not limited to a strict p-value of 0.05), than metrics used in previous analyses, and thus picks up on broader trends. While a 66% QCI is far from decisive, the fact that viral load tended to decrease with increasing propolis score in six different instances suggests that propolis likely has some impact on honey bee viruses.

The mechanism through which propolis impacts viral loads is unclear and could involve additional factors such as mite population (i.e., if propolis reduces mite loads, it may reduce viral transmission by extension). Future studies should examine the relationship between viral loads, mite levels, and propolis deposition across colonies with a wider range of *Varroa* infestation. In our study, the lack of correlation between *Varroa* mite levels and viral loads could be due to the low range in mite loads or it could be suggestive of a direct impact of propolis against these viruses. Regardless of the mechanism, though, our findings build on previous work showing that propolis plays a constitutive role social immunity, where it has a constant, background preventative effect against parasites and pathogens [54]. The question of whether propolis functions as a therapeutic or induced defense against pathogens requires further exploration, particularly in regard to its potential as a treatment against bee disease.

Perhaps related to decreased pathogen pressure, in migratory colonies total bee population was higher in rough boxes colonies compared to control colonies at the end of year one of this study. Although there were no significant differences in total bee populations between rough box and control colonies early in year one, by February of 2020 –ten months into the colony life cycle–migratory rough box colonies were significantly larger than migratory control

colonies, by a margin of nearly two frames of bees plus brood. While colony size cannot be considered a direct measure of colony health, this result is likely significant for beekeepers, particularly those who rent their colonies for crop pollination and must provide growers with a substantial foraging force in order to maximize colony rental costs [55]. Interestingly, throughout year two of the migratory experiment, rough box and control colonies were similar in size though still trended larger. Borba et al. [18] also observed colony-level differences between propolis-rich and propolis-poor colonies during one replicate year but not the other, possibly pointing to context-dependent fluctuations in external factors that support or detract from honey bee health.

The expression of multiple immune genes tended to decrease and stabilize with increasing propolis deposition in both stationary and migratory colonies in August of year one. Decreased immune gene expression in propolis-rich environments is consistent with the results of previous studies [18, 37]. As previously mentioned, in this study, because we used a numerical scoring system to determine the amount of propolis inside the hive (rather than relying on a propolis-rich/propolis-poor binary), we were able to examine the ways in which variation in gene expression changes with respect to propolis score. This analysis revealed a stabilization effect: as propolis deposition increases, gene expression becomes less variable for multiple immune genes. If, following Dawkins [56], a healthy population is characterized by greater uniformity in health-related metrics, then this stabilization effect likely benefits colony health, contributing to hive homeostasis. Indeed, Borba et al. [18] speculated that the modulation of immune system activity might be the most important function of the propolis envelope. The stabilization of immune gene expression may also correspond to the stabilization of the microbiome in propolis-rich environments, which has been observed in previous studies [29, 30]. Full sequencing of the microbiome was beyond the scope of this study. Although it is difficult to draw holistic conclusions based on the abundance of few bacteria, we did note that the expression of multiple bacterial genes was correlated with propolis score. Taken together, results from this study and from previous work indicate that propolis likely plays a strong role in maintaining not only nest environment homeostasis but also social homeostasis, ultimately improving social resilience in the face of stressors [57].

The effects of propolis score on gene expression were seasonal, consistent with previous work [18]. In the migratory operation, propolis deposition had contrasting effects on the expression of immune genes *defensin-1* and *AmEater*, and bacterial gene *B. asteroides* in August and February of one or both years. Seasonal effects of propolis on immune gene expression were also recorded by Borba et al. [18], who noted that the antimicrobial activity of the propolis envelope decreases over the winter, and tends to be lower in the spring. By February of both our study years, the propolis envelope may not have been "fresh," and this could explain why certain trends in gene expression were diluted at this time of year. It is also possible that, since resins from different plants have different antimicrobial properties, the seasonal differences we observed could reflect variation in the resin resources available at different time points [58] Alternatively, the February sample dates may have corresponded to increased immune gene expression due to greater colony stress. In February, colonies were in California for almond pollination. Migratory movement of colonies has been associated with increased viral load [59, 60], increased oxidative stress, and decreased worker bee life span [61]. The presence of EFB symptoms was also notable at this time, and it is possible that colonies were more exposed to pathogens during this period. Turcatto et al. [62] determined that bees fed a propolis-rich diet exhibit increased expression of immune genes when challenged with *E. coli* injection, compared to bees not fed propolis. Thus, it is possible that the increased expression of certain immune genes in bees in propolis-rich rough box environments is reflective of a healthy response to increased environmental stressors. However, myriad interacting factors

influence immune gene expression, so more data is needed to test the effect of environmental stressors on immune gene expression in the presence and absence of propolis. In our study, the collection of gene expression data, and colony health data in general, were unfortunately limited due to COVID-19 travel restrictions.

**Honey production.** Honey production is an important metric for beekeepers operating in a commercial context; we tracked this metric to determine whether the use of rough boxes impacts honey production in any way. There were no differences in honey production across box type in stationary colonies. However, in the migratory operation in year one, rough box colonies did produce less honey than control colonies by an average of 15 kilograms per colony (a 38% decrease). This result is in contrast with findings from other studies, which have found a positive correlation [50, 63] or no correlation [49] between honey production and propolis collection. These contrasting results may point towards the importance of additional factors, such as resource availability and colony size, in shaping nectar and resin foraging dynamics. In our study, migratory beekeepers described year one as "a bad honey year" overall. Indeed, year one honey production was about half that of year two. Notably, differences in honey production across box type were evident in small colonies, but not large colonies (i.e., small rough box colonies produced less honey than small control colonies, but there were no differences in honey production across large control and rough box colonies). It is possible that small rough box colonies produced less honey because foragers were occupied with resin collection, but since these colonies did not bring in significantly more propolis than small control colonies, there is no clear evidence indicating a nectar/resin tradeoff. Further research is required to more fully evaluate the conditions under which this type of tradeoff might emerge. However, in practical terms, a potential resin/nectar tradeoff might only be a short-term concern for beekeepers. Year one rough boxes were reused in year two, so year two colonies were established in boxes already containing propolis, and these colonies demonstrated no differences in honey production across treatments. This might indicate that, once the propolis envelope is established, colonies invest fewer bees in resin foraging, and resin foraging does not detract from honey production. Moving forward, beekeepers may also weigh for themselves the benefits of a health-supportive propolis envelope against the cost of a possible, temporary dip in honey production. It is also possible that, since beekeepers require populous colonies to fulfill pollination contracts, the population boost that rough boxes provide could help balance this calculus.

## Conclusions

Our study demonstrates that using rough boxes to stimulate the construction of a propolis envelope represents an important opportunity to bolster honey bees' natural defenses. Compared to other interventions, using rough boxes to boost propolis collection could be considered an "easy win" because their implementation requires minimal disruption to beekeeping operations and offers measurable benefits to honey bee health in a cost-effective manner. However, the fact that propolis deposition was highly variable even within the rough box treatment suggests that, in addition to modifying box surface texture, further measures should be taken to facilitate the construction of a robust propolis envelope. Some of these measures include fortifying landscapes with resin-producing plants and selecting for bees that engage in resin-hoarding behaviors. Taken together, these actions should contribute substantially to the restoration of the propolis envelope as a natural defense for honey bees.

Importantly, while facilitating the construction of a robust propolis envelope does support bee health, our findings also indicate that propolis is not a silver bullet. Despite clear benefits of propolis to multiple measures of colony health, we observed no differences in survivorship

between box types. This result is not entirely surprising; the restoration of one aspect of social immunity should not be expected to completely counteract the effects of the multiple interacting stressors that threaten bee health within and beyond industrial agriculture systems. This does not diminish the promise of the propolis envelope as a health-supportive tool. Rather, it suggests that rough boxes represent one important intervention to implement in concert with other management, landscape, and systems-level efforts to support honey bee health.

## Supporting information

**S1 Fig. Propolis scoring methods.** Four volunteers were provided reference photos (A) demonstrating what propolis looks like and differentiating wax from propolis. Volunteers were instructed to score photos on a scale from 1–10, based on % coverage of propolis, not on background coloration of the box or comb where it attached (B). Volunteers then used a Google form to fill out a practice survey, which allowed them to view and score ten sample photos. Finally, volunteers completed a full survey, scoring each wall of each box (C). Scores from all four walls, and from all four volunteers were averaged to create to create a composite "propolis score" for each colony.
(PDF)

**S2 Fig. Sample of propolis scoring results.** Control, trap, and rough box hive bodies were evaluated by four volunteers. The black text box at the upper left corner of each photo indicates the score assigned to that photo, according to one volunteer.
(PDF)

**S1 Table. Primers used to quantify the expression of immune, bacterial, and viral genes, as well as genes associated with European foulbrood and two reference genes.**
(XLSX)

**S2 Table. Distribution analysis results characterizing trends in immune gene expression with increasing propolis score for stationary and migratory operations across three sample dates.** Gene expression in seven-day-old bees (stationary) and young bees collected from frames with sealed brood (migratory) was quantified using real-time PCR. Six immune genes were analyzed in stationary colonies (n = 30), and gene expression trends were analyzed at both the apiary and colony level. The same genes, with the exception of *relish*, were analyzed in migratory colonies (n = 102) at the apiary level. A distributional regression model was used to determine the probability that gene expression increases, decreases, stabilizes, or destabilizes with increasing propolis score. The QCI listed refers to the quantile credible interval, as determined by our model, and reflects the widest possible credible interval supporting the indicated trend (not containing zero). Blank lines indicate no trend detected.
(XLSX)

**S3 Table. Distribution analysis results characterizing trends in bacterial gene expression with increasing propolis score for migratory operations across three sample dates.** Gene expression in young bees collected from frames with sealed brood from migratory colonies (n = 102) was quantified using real-time PCR. Six bacterial genes were analyzed at the apiary level. A distributional regression model was used to determine the probability that gene expression increases, decreases, stabilizes, or destabilizes with increasing propolis score. The QCI listed refers to the quantile credible interval, as determined by our model, and reflects the widest possible credible interval supporting the indicated trend (not containing zero). Blank lines indicate no trend detected.
(XLSX)

**S4 Table. Distribution analysis results characterizing trends in viral load with increasing propolis score for migratory operations across three sample dates.** Gene expression in young bees collected from frames with sealed brood from migratory colonies (n = 102) was quantified using real-time PCR. Nine viruses were analyzed at the apiary level. A distributional regression model was used to determine the probability that viral load increases, decreases, stabilizes, or destabilizes with increasing propolis score. The QCI listed refers to the quantile credible interval, as determined by our model, and reflects the widest possible credible interval supporting the indicated trend (not containing zero). Blank lines indicate no trend detected. Viruses with few positive reads were excluded from analysis.
(XLSX)

**S5 Table. Landscape data.** Landscape data was pulled from the USDA-NASS Cropscape database's 2019 Cropland Data Layer (https://nassgeodata.gmu.edu/CropScape/). A circle with a 2.5-mile (4 km) radius was drawn around each apiary (corresponding to honey bees' typical foraging range), and land use statistics were calculated within these defined areas of interest. Land use types were sorted into the following categories: grass and pasture, forest and shrubs, water, herbaceous and woody wetlands, developed, corn and soy, and other crops. Proportional land use was calculated by dividing each category's acreage by the total acreage within the 2.5-mile (4 km) radius. Apiary locations are not disclosed here in order to protect the privacy of the beekeeper who participated in this study. Landscape data for stationary and migratory locations included in this file.
(XLSX)

**S6 Table. Stationary colony field data.** Field data was collected in August of 2019, and included: propolis score, treatment group, brood pattern, brood disease, *Varroa* symptoms, number of frames of bees in the top deep, number of frames of bees in the bottom deep, total number of frames of bees, weight of honey harvested, winter survival, years survived, and inclusion in analysis based on completeness of data.
(XLSX)

**S7 Table. Stationary colony immune gene expression data.** Immune gene expression was measured in 7-day-old bees collected from migratory colonies. qPCR was used to quantify the expression of immune genes *defensin-1*, *abaecin*, *hymenoptaecin*, *AmPPO*, *AmEater*, *and Relish*. Reference genes included ß-actin and Pros54.
(XLSX)

**S8 Table. Migratory colony field data.** Field data was collected in August of 2019, February of 2020, March of 2020, May of 2020, September of 2020, and February of 2021. Not all data were collected for all dates, but data included yard, number of frames of bees, number of frames of brood, total bee population, colony size, honey production, propolis score, whether or not colonies were scored at each date, inclusion in analysis based on completeness of data.
(XLSX)

**S9 Table. Migratory colony brood disease data. Brood disease data (A. apis gene expression and EFB signs) collated for analysis.**
(XLSX)

**S10 Table. Migratory colony honey production data.** Honey production data collated for analysis.
(XLSX)

**S11 Table. Migratory colony immune gene expression data.** Immune gene expression was measured in young bees collected from migratory colonies. qPCR was used to quantify the expression of immune genes *defensin-1*, *abaecin*, *hymenoptaecin*, *AmPPO*, and *AmEater* Reference genes included ß-actin and Pros54.
(XLSX)

**S12 Table. Migratory colony bacterial gene expression data.** Bacterial gene expression was measured in young bees collected from migratory colonies. qPCR was used to quantify the expression of bacteria *Bartonella apis*, *Bifidobacterium asteroides*, *Lactobacillus* Firm-4 phylotype, *Lactobacillus* Firm-5 phylotype, *Snodgrassella alvi*, and *UniBact*, a primer coding for a universal bacterial gene sequence. Reference genes included ß-actin and Pros54.
(XLSX)

**S13 Table. Migratory colony virus data.** Gene expression in young bees collected from migratory colonies and the viral loads were quanitified for the following viruses: acute bee paralysis virus (ABPV), black queen cell virus (BQCV), chronic bee paralysis virus (CBPV), deformed wing virus A (DWV-A), deformed wing virus B (DWV-B), Israeli acute paralysis virus (IAPV), Kashmir bee virus (KBV), Lake Sinai virus 1 (LSV-1), and Lake Sinai virus 2 (LSV-2); genes associated with European foulbrood; as well as reference genes p*ros54* and *ß-actin*.
(XLSX)

## Acknowledgments

We would like to thank Minnesota beekeeper Christian Dahm of the Propolis Hive Company for developing and providing the rough boxes used in this experiment. We would also like to thank Gary Reuter, Yuuki Metreaud, Héctor Morales Urbina, Jenny Warner, and Judy Griesedieck at the University of Minnesota for helping realize the stationary component of this experiment. We appreciate the efforts of Bob Cox, David Dodge, Victor Rainey, RaeDiance Fuller, Sarah Lang, Hunter Martin, Natalie Martin, Christin Moreau, Nathan Egnew, Allyson Martin, and Mandy Frake in assisting with field and laboratory aspects of the migratory experiment. We also thank the cooperating large-scale beekeeper and their outstanding employees, without whom this work would not have been possible. Finally, we are grateful to the Bee Informed Partnership and Nelson Williams for helping sample migratory colonies in almonds during February 2021.

Mention of trade names or commercial products in this publication is solely for the purpose of providing specific information and does not imply recommendation or endorsement by the U.S. Department of Agriculture. USDA is an equal opportunity provider and employer.

## Author Contributions

**Conceptualization:** Michael Simone-Finstrom, Frank Rinkevich, Marla Spivak.

**Formal analysis:** Maggie Shanahan, Philip Tokarz, Frank Rinkevich, Quentin D. Read.

**Funding acquisition:** Michael Simone-Finstrom, Frank Rinkevich, Marla Spivak.

**Investigation:** Maggie Shanahan, Michael Simone-Finstrom, Philip Tokarz, Frank Rinkevich, Marla Spivak.

**Methodology:** Maggie Shanahan, Michael Simone-Finstrom, Philip Tokarz, Frank Rinkevich, Quentin D. Read, Marla Spivak.

**Project administration:** Michael Simone-Finstrom, Marla Spivak.

**Resources:** Michael Simone-Finstrom, Marla Spivak.

**Supervision:** Michael Simone-Finstrom.

**Writing – original draft:** Maggie Shanahan, Quentin D. Read.

**Writing – review & editing:** Maggie Shanahan, Michael Simone-Finstrom, Frank Rinkevich, Quentin D. Read, Marla Spivak.

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
