## [Decision Letter · Decision Letter 0]

5 Jun 2023

PONE-D-23-10474Thinking inside the box: Restoring the propolis envelope facilitates honey bee social immunityPLOS ONE

Dear Dr. Shanahan,

Thank you for submitting your manuscript to PLOS ONE. I have been finally able to secure two qualified reviews. My apology for the delay. The reviewers generally advise that this study is valuable and I agree. With the correct moderation of language to guide the readers through this complex study, I see no major issues. However, I decided to ask for a major revision (instead of a minor revision) due to the sheer number of comments and suggestions.

We look forward to receiving your revised manuscript.

Kind regards,

Olav Rueppell

Academic Editor

PLOS ONE

Journal Requirements:

Reviewers' comments:

Reviewer's Responses to Questions

**Comments to the Author**

1. Is the manuscript technically sound, and do the data support the conclusions?

Reviewer #1: Yes

Reviewer #2: Yes

2. Has the statistical analysis been performed appropriately and rigorously? 

Reviewer #1: Yes

Reviewer #2: Yes

3. Have the authors made all data underlying the findings in their manuscript fully available?

Reviewer #1: Yes

Reviewer #2: Yes

4. Is the manuscript presented in an intelligible fashion and written in standard English?

Reviewer #1: Yes

Reviewer #2: Yes

5. Review Comments to the Author

Reviewer #1: The authors addressed a form of social immune response that seems relevant for honey bees, propolis envelope. Giving hives with grooves to encourage a propolis envelope seems like a practical way in which beekeepers could encourage a social defense mechanism to limit pathogens. I understand there are few studies on this topic, so all the new information is very valuable. I thank the authors for the transparency about the limitations of the study due to Covid-19 pandemic. I believe the topic is relevant for beekeeping, and also to understand more about the mechanisms of social immunity in honey bees. Although some analyses are limited, (e.g. limited statistical comparisons between stationary and migratory colonies), the experimental design and the statistical analyses seem appropriate. Also, the conclusions are realistic and within the scope of the experimental design and analyses. I hope the authors and the Editor find my observations helpful.

L91-92 I suggest following the nomenclature for viruses by the ICTV < https://ictv.global/faq/names>

L417 I’m just wondering if there is a rationale behind the size of the grooves.

L317 Is there a reason why stationary colonies were only scored in August? Would it had been better to record during the same (or similar) dates for both groups (stationary and migratory)?

L350 The degree symbol is missing before the “C” of Celsius.

L367 Is there a reason why the stationary bees were not assessed for Borrtonella apis, Bifidobacterium asteroides, Lactobacillus Firm 4 phylotype, Lactobacillus Firm-5 phylotype, Snodgrassella alvi, and UniBact, and viruses mentioned in L369-371?

L408 Why were the samples with Ct values <23.5 discarded? Is it related to a standard curve to optimize the performance of reference and target genes? It seems like 23.5 is a high threshold.

L780. I understand that full sequencing of the microbiome was not the scope of this study, and the results are relevant to determine if propolis has an effect on the microbiome. Conclusions based on the abundance of few bacteria are difficult (or limited).

General comment. It would be interesting to determine which immune pathways are more likely to be triggered by the presence of a propolis envelope, but I understand that the authors used immune related genes as markers. Also, to determine the composition of the propolis collected by the bees in the different locations and at different timepoints would have been interesting (along with gene expression analysis and identification of pathogens, this might change over the season).

L1110 and L302 Was the variability between observes estimated for the propolis scoring method?

No comments on the Supplementary files.

Reviewer #2: Thinking inside the box: Restoring the propolis envelope facilitates honey bee social immunity

PONE-D-23-10474

Reviewer’ comments

This manuscript addressed the effects of different hive boxes and beekeeping systems on the propolis production in honey bee colonies. I think the findings will be of interest to many beekeepers and researchers. However, in this unbalance study several variables were evaluated such as:

stationary beekeeping vs migratory beekeeping;

3 treatments (in stationary) vs 2 treatments (in migratory);

38 colonies (in stationary) vs 120 colonies (in migratory);

1-year study (in stationary) vs 2-year study (in migratory);

Saskatraz queens (in stationary) vs queens from local breeders (in migratory);

Single landscapes (in stationary) vs different landscapes (in migratory);

a scale was used to measure bee population (in stationary) vs a standard method (in migratory);

4 mounts propolis collection (in stationary) vs 2 years (in migratory);

20 bee samples for molecular analysis (in stationary) vs 300 bees (in migratory);

RNA extraction using the reagent TRIzol (in stationary) vs the Maxwell RSC kit (in migratory);

No pathogen (in stationary) vs EFB and viruses (in migratory);

Therefore, manuscript needs major revisions to be smooth for readers to follow up results and discussion.

Minor comments:

L 54: (Apis mellifera L.)

L84-85: Ascosphaera apis (Maasen ex Claussen)

L87: Varroa mites (Varroa destructor, Anderson and Trueman)

L93: Nosema ceranae Fries

L142-145: Please show the dimension of each type of boxes

L152: …. supporting colony health (Reference?).

L 168: The stationary component of this study was conducted in 2019-2020 at ....

L 169: The total is showed n=38, however in lines 171-172 this number is 32.

L 171: The section 2.1 has not been indicated in the manuscript.

L176: The migratory component of this experiment was conducted for two years in 2109-2020 and 2020-201……

L179: In the migratory system queens were provided from queen breeders. Did the difference between source of queens (migratory vs stationary) effect the results? Please cite information of queen breeders in migratory section.

L186: Reference for measuring brood and adult bee.

L188: Please explain how did you standardize colonies based on the brood and bee population.

L188-189: Explain reasons that the propolis trap treatment was excluded from the migratory.

L194: Was the residue of propolis from the last year measured and included in the results of the second year?

L200: a second hive body (brood box) ….

L203: Please show the miticide and treatment details that was used in Varroa treatment.

L201: References for the beekeeper’s standard practices.

L213-214: Why did you add 34 extra colonies into the experiment? Did you include the data from these 34 colonies in the results?

L217: Please clarify how did you determine that environment was high-propolis.

L222: Was pollen patty not used for stationary? Cite the manufacture and brand of pollen supplement, if it was not homemade.

L226: Cite the miticide that was used for Varroa.

L229: When hives were moved into the almond orchards in early February 2020, were they single or doubled?

L139: a 2.5-mile (= ?? Km)

L241-242: Please cite reference(s) for “Land use types were sorted into the following categories:”.

L251: “In August, frames of bees were counted for both the top and bottom hive body.” How did you calculate the total bee population?

L254: The scale (poor, fair and good) is usually used by beekeepers not for a scientific study.

L256-257: Did you weigh empty super boxes before honey season or after honey harvest?

L258: The section 2.5 is not found in the paper.

L259: Please clarify the reason that healthy colonies were selected based on brood frames only, not on adult bees or both.

L267-268: In the stationary, a traditional scale from 1 to 3 (1 = poor, 2 = fair, 3 = good) was used to measure the bee pollution, however, for the migratory a standard method was used. This is confusing that how two different methods were applied.

L269: The hives were treated for Varroa control. When did you determine the mite level?

L276: The section 2.5 is not showed in the paper.

L283: European Foulbrood (EFB) (Melissococcus plutonius (ex White 1912)

L258: Cite reference(s) for the protocol.

L 291-292: Please explain why did you use different scales for brood pattern in migratory, where another method was used for stationary. Cite the reference for the rank. In both migratory and stationary, did beekeepers measure the brood and bee population? If so, please show it in the M&M.

L297-298: What does it mean that propolis deposition was assessed within one week? In stationary it looks took four months.

L302: Briefly explain the method of Hodges et al.

L322: The propolis deposition was measured for a certain time of four months in stationary colonies, where it did 3 times for a 2-year in the migratory. It is complicated to compare the results of two different set of data that were collected with different set point of sampling.

L331: Please show the manufacture info for the enamel paint.

L336: Section 2.5.2. is not found.

L337: In the stationary section, 20 bees were sampled however 300 bees for migratory. How did this big difference happen in the bee number?

L362: RNA was extracted from 20 bees in stationary, where a pool of 30 bees was used in migratory! Please clarify the reason that you did not follow the same method.

L368-373: For migratory, the expression of a few bacteria and viruses were quantified, but not for stationary!

L391: number of frames of adult bees,….

L392: The hives were treated for control Varroa. Please indicate when the mite load was measured.

L396-397: Please clarify what variables were analyzed using the mixed-effects models and for what beekeeping system.

L432: To be consist in the paper, first report the results of stationary then migratory.

L439: Positive correlation for stationary or migratory?

L454-456: Did you separately compare means of the rough boxes with trap and control? What is difference between trap and control?

L458-459: Show the full statistic for both P values.

L466-467: Is it possible to present means for overall propolis collection in each treatment?

L482: For migratory, frames of bees were counted for all colonies at all sampling dates.

L483-484: It is recommended to determine where you exactly combined the brood and adult bees as bee population.

L485: There were no significant difference ….. . Please show statistic.

L491: a non-significate increase ….. . Please show statistic.

L 509, 513 and 516: Please show statistic.

L 517-518: Was decreasing in the defensin-1 expression significantly differed? Show statistic.

L520-525: Indicate statistic for all increasing/decreasing expressions.

L543-545: Indicate statistic. Both sentence looks have the same mean for different genes. please combine both sentence.

L554-556 and 587: Show statistic.

L590: The expression of M. plutonius …

L595-597: If colonies were treated for Varroa, how does the reduction in infestation associate with the propolis collection?

L598-599: It looks that changes in the viral load may correlated with mite levels. Please discuss it.

L618-619: Explain why stationary colonies were wintered without enough food.

L626: Show statistic of results for honey production.

L637: Indicate the statistic for no effect of box type on honey production.

L669: Results did not support it. Results only compared rough boxes with control for migratory!

L673-676: Does difference between grooves sizes may make difference in propolis deposition?

L676: Overall, I suggest to show “mean± se/sd” for variables/amount, when you say more propolis deposited, more honey production or less survival. This helps readers to understand and follow the concept.

L680: How much increase in the propolis deposition was measured over time?

L691-692: Please explain the high or low plant biodiversity mean along with quantification/qualification and statistic.

L701: Could you give a statement on how different bee genotypes affect the results?

L705: Is any genotype bee stock available in market for propolis production in the USA?

L712: Was a marginal reduction in Varroa load associated with propolis or miticide treatment?

L714: 0.5 mites per 100 bees is low. How much is the seasonal mite threshold?

L718: Is it meaning the propolis kills mites? In this study you did not measure the mite mortality, while mite mean abundance was evaluated.

L722-726: This sentence is not relevant with the paragraph.

L728-729: M. plutonius –

L729: Delete “– the causative agent of European foulbrood (EFB)”. This was already cited.

L730-731: To state like this, you should artificiality infect bees with pathogen, then see if propolis able to reduce the EFB infection in compared with the control.

L736: The reduction in viral load may affected by other factors or interaction between biological aspect of bee colonies, mite population and propolis collection.

L752: Propolis function can also be counted as an improvement in the immune system.

L755: “an increase in total bee …”. this may have related to bee genotype and environmental factors in compared with stationary.

L770-771: This study did not quantify the propolis. Quantification means that you measure the weigh of collected propolis. However, you evaluate propolis based on a scale not weight.

L785-786: … with previous work (references).

L786: In migratory, propolis deposition had contrasting effects on ….

L790-792: Different collected resin may affect the antimicrobial property of seasonal propolis.

L794: Did bees get any stress from agro-pesticides in the almond pollination time?

L813: 33 pounds per colony (= ??? kg/ colony)

L818: “a bad honey year”. Bad year in comparing with what? In this study all colonies were compared with control. So if the year had a low honey season, it was equal for all treatments and control.

L828: Year one rough boxes were reused in year two with residue of propolis from last year, …

6. PLOS authors have the option to publish the peer review history of their article (what does this mean?). If published, this will include your full peer review and any attached files.

Reviewer #1: No

Reviewer #2: No

---

## [Author Response · Author response to Decision Letter 0]

10 Aug 2023

[point-by-point responses also available in "cover letter" document submitted alongside revisions, with formatting that may facilitate readability]

Reviewer #1: 

The authors addressed a form of social immune response that seems relevant for honey bees, propolis envelope. Giving hives with grooves to encourage a propolis envelope seems like a practical way in which beekeepers could encourage a social defense mechanism to limit pathogens. I understand there are few studies on this topic, so all the new information is very valuable. I thank the authors for the transparency about the limitations of the study due to Covid-19 pandemic. I believe the topic is relevant for beekeeping, and also to understand more about the mechanisms of social immunity in honey bees. Although some analyses are limited, (e.g. limited statistical comparisons between stationary and migratory colonies), the experimental design and the statistical analyses seem appropriate. Also, the conclusions are realistic and within the scope of the experimental design and analyses. I hope the authors and the Editor find my observations helpful.

Thank you for this helpful feedback.

L91-92 I suggest following the nomenclature for viruses by the ICTV (https://ictv.global/faq/names). 

Noted, thank you! Changes made.

L417 I’m just wondering if there is a rationale behind the size of the grooves. 

Good question. Grooves were designed to imitate the inner surface of a tree cavity. They were designed to be thick enough to allow bees to reach down into them (and thus minimize the chance of wax moth or small hive beetle establishing beyond the bees’ reach), and thin enough so that an individual bee could still perceive the gap with her antennae. These particular boxes were made by a specific supplier using proprietary technology; at the time of this study, theirs was the only company making boxes like this. 

L317 Is there a reason why stationary colonies were only scored in August? Would it had been better to record during the same (or similar) dates for both groups (stationary and migratory)? 

Thanks for calling our attention to this point. The stationary component of this experiment was meant to serve as a proof-of-concept experiment to compare an established method (use of propolis traps) to a new method (rough boxes). The migratory component was designed as a separate experiment, where our goal was to test to the use of rough boxes in a migratory beekeeping context. We see now that this distinction was not communicated as clearly as it could have been in the original manuscript. We changed the language in the final paragraph of the introduction (L129-134) and the first two paragraphs of the methods section (L147-147; L156-165) to clarify that the stationary and migratory components of the experiment were separate studies not designed to be compared to each other. 

Experimental design aside, it would have been interesting to measure propolis score in stationary colonies at multiple points during the year. However, we were able to score migratory colonies in February because they were in California (warm climate) and this is a key point in the commercial beekeeping cycle (almond pollination). Our stationary colonies were in Minnesota in February (cold climate), and we were unable to open colonies at this time without exposing them to the cold. 

L350 The degree symbol is missing before the “C” of Celsius. Good catch, change made.

L367 Is there a reason why the stationary bees were not assessed for Borrtonella apis, Bifidobacterium asteroides, Lactobacillus Firm 4 phylotype, Lactobacillus Firm-5 phylotype, Snodgrassella alvi, and UniBact, and viruses mentioned in L369-371? We had to limit the number of genes we tested for due to the limited amount of RNA we were able to extract from the bees we collected from stationary colonies.

L408 Why were the samples with Ct values <23.5 discarded? Is it related to a standard curve to optimize the performance of reference and target genes? It seems like 23.5 is a high threshold. Good question. Reference genes with Ct values below 23.5 and above 30 cycles were statistically identified as outliers. We excluded 7/225 data points from the stationary data set and 6/243 data points from the migratory data set, so it was a marginal number of samples. We added an additional sentence (L499-L500) to clarify that there were different thresholds for exclusion for the different data sets (because samples were collected slightly differently, and run at different times).

L780. I understand that full sequencing of the microbiome was not the scope of this study, and the results are relevant to determine if propolis has an effect on the microbiome. Conclusions based on the abundance of few bacteria are difficult (or limited). This is a valid point, and worth emphasizing in the manuscript. We viewed this portion of the experiment as something of an exploratory analysis following up on sequencing-based efforts that were previously published (Saelao et al. and Dalenberg et al.), which similarly found effects largely based on the core bacteria examined here. The results presented here provide avenues for hypothesis development regarding particular taxa. But, we appreciate your point and have added “Although it is difficult to draw holistic conclusions based on the abundance of few bacteria” to the discussion section (L948-949).

General comment. It would be interesting to determine which immune pathways are more likely to be triggered by the presence of a propolis envelope, but I understand that the authors used immune related genes as markers. Also, to determine the composition of the propolis collected by the bees in the different locations and at different timepoints would have been interesting (along with gene expression analysis and identification of pathogens, this might change over the season). Yes, this would be interesting! We hope future work will explore both these avenues.

L1110 and L302 Was the variability between observes estimated for the propolis scoring method? Good catch. Added: “A Fleiss kappa test indicated a fair amount of agreement between evaluators (Kappa = 0.25).”

No comments on the Supplementary files.

Reviewer #2: 

This manuscript addressed the effects of different hive boxes and beekeeping systems on the propolis production in honey bee colonies. I think the findings will be of interest to many beekeepers and researchers. However, in this unbalance study several variables were evaluated such as:

stationary beekeeping vs migratory beekeeping;

3 treatments (in stationary) vs 2 treatments (in migratory);

38 colonies (in stationary) vs 120 colonies (in migratory);

1-year study (in stationary) vs 2-year study (in migratory);

Saskatraz queens (in stationary) vs queens from local breeders (in migratory);

Single landscapes (in stationary) vs different landscapes (in migratory);

a scale was used to measure bee population (in stationary) vs a standard method (in migratory);

4 mounts propolis collection (in stationary) vs 2 years (in migratory);

20 bee samples for molecular analysis (in stationary) vs 300 bees (in migratory);

RNA extraction using the reagent TRIzol (in stationary) vs the Maxwell RSC kit (in migratory);

No pathogen (in stationary) vs EFB and viruses (in migratory);

Therefore, manuscript needs major revisions to be smooth for readers to follow up results and discussion.

Thank you for pointing this out! We see now that the description of the experimental set up in the original manuscript was unclear. We did not set up this experiment to make comparisons between stationary and migratory colonies. Rather, the stationary component was meant to serve as a proof-of-concept experiment, comparing the use of rough boxes to a strategy previously proven to stimulate the construction of a propolis envelope and support colony health. The migratory component was meant to test whether rough boxes can support colony health within large-scale commercial beekeeping operations. Because propolis deposition in rough boxes turned out to be so different across contexts (stationary vs. migratory) we did end up making comparisons of propolis deposition and of landscape composition. For propolis deposition comparisons, although the number of colonies in stationary/migratory is not balanced, we only made within-treatment comparisons (e.g., stationary rough boxes vs. migratory rough boxes). We did not compare other variables across contexts (stationary vs. migratory).

We changed the language in the last paragraph of the introduction section and the first two paragraphs of the methods section to help clarify the comparison we hope to make here, and we hope this clears up this point of confusion.

Minor comments:

L 54: (Apis mellifera L.) Change implemented

L84-85: Ascosphaera apis (Maasen ex Claussen). Change implemented

L87: Varroa mites (Varroa destructor, Anderson and Trueman). Change implemented

L93: Nosema ceranae Fries. Change implemented

L142-145: Please show the dimension of each type of boxes. Added “Langstroth-size deep hive boxes” to each treatment descriptor to demonstrate that the standard dimensions for each box type.

L152: …. supporting colony health (Reference?). Added Borba 2015 reference

L 168: The stationary component of this study was conducted in 2019-2020 at .... change implemented

L 169: The total is showed n=38, however in lines 171-172 this number is 32. Good catch! Change implemented

L 171: The section 2.1 has not been indicated in the manuscript. Changed to: “described in Figure 1”

L176: The migratory component of this experiment was conducted for two years in 2109-2020 and 2020-201…… Changed to: “The migratory component of this experiment was conducted over the course of two years from 2019-2021”

L179: In the migratory system queens were provided from queen breeders. Did the difference between source of queens (migratory vs stationary) effect the results? Good question. This is discussed beginning on line 697 of the original submission: “Genetic differences may have also contributed to the variation we observed in propolis score, both between migratory and stationary yards, and between colonies in the same box type, in the same yard. Propensity for propolis collection is a highly heritable trait (coefficient of heritability = 0.87 [49]), and selection efforts can yield high-propolis colonies [50]. Different honey bee stocks were used in the stationary and migratory study, which, along with landscape differences, may have contributed to the variation in propolis deposition across contexts.” Please cite information of queen breeders in migratory section. This information is included on L179 of original manuscript (L197 revised manuscript). We added that these are Italian bees: “Queens were grafted from Italian breeder queens selected from within the operation.”

L186: Reference for measuring brood and adult bee. Reference added

L188: Please explain how did you standardize colonies based on the brood and bee population. Made this change: “frames were moved between colonies to ensure similar amounts of sealed brood and adult bee population”

L188-189: Explain reasons that the propolis trap treatment was excluded from the migratory. Added: “Propolis trap colonies were not included in the migratory component of this study due to the cost of installing propolis traps at scale, and the difficulties these present for beekeepers manipulating frames.”

L194: Was the residue of propolis from the last year measured and included in the results of the second year? Propolis from year one was not removed from boxes, nor was propolis scored when new colonies were established in used rough boxes. Propolis scores in year two represent the propolis collected from year one and year two colonies. 

L200: a second hive body (brood box) …. Change made

L203: Please show the miticide and treatment details that was used in Varroa treatment. Added: “…with Formic Pro® (NOD Apiary Products, Ontario, Canada) according to the labeled instructions: 1 was applied strip between brood chambers for 10 days and then replaced with another strip for another 10 days after which the strip was removed.

L201: References for the beekeeper’s standard practices. Changed to: “All management of colonies followed the cooperating beekeeper’s proprietary standard practices as described below.”

L213-214: Why did you add 34 extra colonies into the experiment? Did you include the data from these 34 colonies in the results? These 34 colonies were not included in the experiment (“as well as 34 colonies unrelated to the study”). They were present because the migratory component of this study was conducted in collaboration with a migratory beekeeper, who generously allowed for a portion of his colonies to participate in the study. 64 colonies is a standard apiary size that allows for consistent and predictable logistics.

L217: Please clarify how did you determine that environment was high-propolis. Changed to: “Additional rough boxes were provided to the beekeeper to add to the colonies after transport from Mississippi to South Dakota so that the rough box treatment surrounded brood chamber, consisting of the bottom two boxes of each experimental colony.”

L222: Was pollen patty not used for stationary? Cite the manufacture and brand of pollen supplement, if it was not homemade. The pollen patty (with manufacturer and part number) used for the stationary colonies is referenced on L199 of the original manuscript (L228 revised), but we’d failed to include the manufacture and brand. Thanks for catching this! Added “a patty of Ultra Bee High Protein…” Added additional detail to the migratory pollen patty reference: “500g soya-based protein supplement patty manufactured in-house according to the beekeeper’s proprietary formula.”

L226: Cite the miticide that was used for Varroa. We have included miticide information for the stationary colonies but are not able to disclose the collaborating beekeepers’ propriety practices.

L229: When hives were moved into the almond orchards in early February 2020, were they single or doubled? Doubles; change implemented

L139: a 2.5-mile (= ?? Km) Change implemented

L241-242: Please cite reference(s) for “Land use types were sorted into the following categories:”. We don’t actually have a source supporting this choice. We chose to combine multiple crops into a single category, distinguish corn/soy crops (a major monoculture in the Midwest) from other crops, and collapse a couple of the categories that Cropscape created (e.g., herbaceous wetland + woody wetland) to make the data more legible. We will make this data available along with the rest of our data so that readers have access to both the original categories as well as the combined categories.

L251: “In August, frames of bees were counted for both the top and bottom hive body.” How did you calculate the total bee population? Total bee population was calculated for migratory colonies, but not for stationary colonies. For stationary colonies, we only counted frames of bees, due to limited time. Clarified in migratory section: “Adult bee population (frames of bees) and amount of sealed brood were visually estimated using standard methods [35], and total bee population was calculated by adding frames of bees and frames of brood.”

L254: The scale (poor, fair and good) is usually used by beekeepers not for a scientific study. This is true! And we can remove this sentence from the manuscript if necessary. For now, changed to: “brood pattern was evaluated using a scale commonly used by beekeepers (1 = poor, 2 = fair, 3 = good).”

L256-257: Did you weigh empty super boxes before honey season or after honey harvest? Revised this sentence: “Honey supers (i.e., the boxes located at the top of the hive where the bees store excess honey) were removed and weighed, and the weight of an average empty honey super (calculated by averaging the weight of ten empty supers) was subtracted from each to determine the approximate weight of the honey inside.

L258: The section 2.5 is not found in the paper. Good catch! Change made

L259: Please clarify the reason that healthy colonies were selected based on brood frames only, not on adult bees or both. This was a typo, it should read frames of bees.

L267-268: In the stationary, a traditional scale from 1 to 3 (1 = poor, 2 = fair, 3 = good) was used to measure the bee pollution, however, for the migratory a standard method was used. This is confusing that how two different methods were applied. We agree that it would have been ideal to use standard methods across contexts. Since bee populations were compared within, rather than across, contexts (stationary/migratory), we hope this error is not too disruptive.

L269: The hives were treated for Varroa control. When did you determine the mite level? Changed to: “Colonies were inspected and sampled immediately after honey harvest and then treated for Varroa mites following proprietary practices.”

L276: The section 2.5 is not showed in the paper. Change made

L283: European Foulbrood (EFB) (Melissococcus plutonius (ex White 1912) Change made

L258: Cite reference(s) for the protocol. Changed to “using the visual scoring system described below” and cited references in the section where this protocol is described in detail.

L 291-292: Please explain why did you use different scales for brood pattern in migratory, where another method was used for stationary. Cite the reference for the rank. In both migratory and stationary, did beekeepers measure the brood and bee population? If so, please show it in the M&M. We agree that it would have been ideal to use the same scale for brood pattern in both stationary and migratory colonies. Changed to: “Queen status, frames of brood, frames of bees, brood pattern (ranked 1-5 with 1 being poor and 5 being a solid brood pattern), were measured during almond pollination in California in February of 2021 by contracted members of the Bee Informed Partnership’s Tech Transfer Team. Brood amount was scored on a 5-point scale compared to the amount of brood area following their standard protocol.”

L297-298: What does it mean that propolis deposition was assessed within one week? In stationary it looks took four months. The sentence “In year one, for both stationary and migratory colonies, propolis deposition was assessed within one week of collecting bee samples for gene expression analysis” refers to the fact that propolis deposition was assessed close to the time of sample collection for gene expression analysis. This is important because propolis deposition changes over time, and if we hope to understand the relationship between gene expression and the amount of propolis present inside a hive, these measures must be taken as close together as possible. In the stationary and migratory sections that follow, we specify that “Propolis deposition was scored in August of 2019, after colonies had been established for four months.” And “Propolis deposition was scored at three time points…” respectively. Please let us know if this point requires further clarification.

L302: Briefly explain the method of Hodges et al. Moved Hodges reference to the beginning of the paragraph and used a colon to indicate that the methods that follow were adapted from Hodges et al. 

L322: The propolis deposition was measured for a certain time of four months in stationary colonies, where it did 3 times for a 2-year in the migratory. It is complicated to compare the results of two different set of data that were collected with different set point of sampling. The main points of comparison for propolis collection are between treatments (box type) and between time points, but within-context (e.g., within stationary, or within migratory). Since we did observe such stark differences in propolis score between stationary and migratory rough box colonies in August 2019, we added this comparison. It is true that this comparison is not balanced (more migratory colonies than stationary colonies). However, we still feel that it is useful to note. 

L331: Please show the manufacture info for the enamel paint. Change implemented

L336: Section 2.5.2. is not found. Change made

L337: In the stationary section, 20 bees were sampled however 300 bees for migratory. How did this big difference happen in the bee number? In stationary colonies, we paint-marked newly emerged bees to ensure the bees we collected were seven days old. This technique is extremely time-consuming, and would have been logistically impossible to replicate in a migratory context, with over a hundred colonies spread across multiple apiaries. Added “Due to the large number of migratory colonies involved in this experiment, paint-marking and collecting 7-day-old bees from migratory colonies was not practical. Therefore…”

L362: RNA was extracted from 20 bees in stationary, where a pool of 30 bees was used in migratory! Please clarify the reason that you did not follow the same method. For the stationary experiment, bees were individually analyzed, as there were only a total of ~30 colonies. Given there were 120 colonies in the migratory experiment, it was more cost-effective to pool bees. Since the stationary and migratory portions of the study were separate experiments, we feel these differences should not impact the interpretation of results.

L368-373: For migratory, the expression of a few bacteria and viruses were quantified, but not for stationary! We had a smaller volume of RNA to work with from the stationary colonies and chose to prioritize immune gene expression, in order to compare to a previous study that used propolis traps to stimulate propolis deposition.

L391: number of frames of adult bees,…. Change implemented

L392: The hives were treated for control Varroa. Please indicate when the mite load was measured. See comment for line 269. 

L396-397: Please clarify what variables were analyzed using the mixed-effects models and for what beekeeping system. We divided this section into two separate paragraphs to more clearly differentiate between the stationary and migratory analyses, and we listed the variables used in our mixed-effects models.

L432: To be consist in the paper, first report the results of stationary then migratory. Thanks for catching this! Change implemented

L439: Positive correlation for stationary or migratory? Changed to “Our simple linear model indicated that across stationary and migratory landscapes, propolis score was positively correlated (r = 0.94, p = 0.02) with percent cover of herbaceous and woody wetlands and forest and shrubs, landscapes likely rich in resin resources.”

L454-456: Did you separately compare means of the rough boxes with trap and control? What is difference between trap and control? Added: “Propolis score was also significantly greater in propolis trap boxes compared to control boxes (t(27) = 11.8, p < 0.001).”

L458-459: Show the full statistic for both P values. Changed to: In migratory colonies, our mixed-effects model indicated that propolis deposition was significantly affected by both box type (t(167) = -9.1; p < 0.0001) and the interaction between box type and sample date (t(167) = 9.3; p < 0.0001).

L466-467: Is it possible to present means for overall propolis collection in each treatment? Yes, good idea. Changed to: “Propolis deposition was higher in rough box stationary colonies (propolis score 7.5; SE = 0.2) compared to rough box migratory colonies (propolis score 3.2; SE = 0.2) in August of 2019 when all colonies were evaluated (F(1,102) = 72.5, p < 0.0001).

L482: For migratory, frames of bees were counted for all colonies at all sampling dates. Changed to: “For both migratory and stationary colonies, frames of bees were counted for all colonies at all sampling dates; for migratory colonies frames of brood were also counted at multiple time points during both years of the experiment.”

L483-484: It is recommended to determine where you exactly combined the brood and adult bees as bee population. Changed to: “Where possible (August of 2019, February of 2020, February of 2021), we combined frames of brood and frames of bees to calculate “total bee population.”

L485: There were no significant difference ….. . Please show statistic. Changed to: There were no significant differences in the number of frames of bees across treatment for the stationary colonies (Fig. 5; F(2,32) = 0.07, p = 0.9).

L491: a non-significate increase ….. . Please show statistic. Changed to: We observed a non-significant increase in total bee population in rough box colonies (F(1,53) = 1.1, p = 0.3).

The following comments concern statistical reporting the Bayesian distributional analyses we conducted to discern patterns in immune gene expression, and will be addressed, as a group, following comment L554-556.

L 509, 513 and 516: Please show statistic.

L 517-518: Was decreasing in the defensin-1 expression significantly differed? Show statistic.

L520-525: Indicate statistic for all increasing/decreasing expressions.

L543-545: Indicate statistic. Both sentence looks have the same mean for different genes. please combine both sentence. The distinction between “some evidence” and “moderate evidence” may be more apparent if we report the QCI in-text (see below). We’re definitely open to making this change – see below.

L554-556 and 587: Show statistic.

Thank you for your attention to detail! Because this is a Bayesian analysis, we report median posterior parameter estimates and construct credible intervals using the central 95% quantiles of the posterior distribution. Null hypothesis significance tests are not used. The posterior estimates of the rate of change of mean and standard deviation of gene expression with respect to propolis score, and their credible intervals, are what our statistical inference is based on. In the original manuscript, we reported all quantile credible intervals in a table (Table S2) to facilitate readability. We are happy to move these numbers to the text if this is preferred. Below we have included the immune gene expression results, with QCI reported in-text. Please let us know if this is the preferred format:

For the August 2019 sample date in both stationary (n = 30) and migratory (n = 102) colonies, immune gene expression tended to decrease with increasing propolis score. In stationary colonies, our distribution model provided strong evidence for a negative correlation between propolis score and defensin-1 expression (Median: -0.105, 95% QCI: [-0.194, -0.012]). This model also provided some evidence that relish (Median: -0.037, 66% QCI: [-0.06, -0.014]), hymenoptaecin (Median: -0.219, 66% QCI: [-0.35, -0.088]), and AmEater expression (Median: -0.049. 66% QCI: [-0.094, -0.002]) decreased with increasing propolis score. However, propolis score was positively correlated with AmPPO expression (Median: 0.068, 90% QCI: [0.01, 0.124]). 

In migratory colonies, our distribution model provided moderate evidence that defensin-1 expression (Median: -0.316, 90% QCI: [-0.6, -0.045]) tended to decrease with increasing propolis score, and some evidence that abaecin expression (Median: -0.071, 66% QCI: [-0.123, -0.021]) tended to decrease with increasing propolis score. 

Our distribution model also provided some evidence that immune gene expression stabilized as propolis score increased. Variation in hymenoptaecin (Median: -0.092, 90% QCI: [-0.184, -0.003]), AmEater (Median: -0.097, 66% QCI: [-0.16, -0.033]) and abaecin (Median: -0.090, 66% QCI: [-0.146, -0.037]) expression tended to decrease with increasing propolis score in stationary colonies, as did variation in abaecin expression (Median: -0.072, 66% QCI: [-0.116, -0.029) in migratory colonies. In contrast, our distribution model suggested that AmEater expression (Median: 0.097, 90% QCI: [0.006, 0.187]) tended to destabilize with increasing propolis score. 

When we compared gene expression in individual bees from stationary colonies, we found convincing evidence that variation in relish expression decreased with increasing propolis score at the colony level (Median: -0.019, 99% QCI: [-0.036, -0.002]), and some evidence that variation in defensin-1 expression decreased with increasing propolis score at the colony level (Median: -0.013, 66% QCI: [-0.023, -0.002]) (Fig. 7).

In migratory colonies, where bees were sampled for immune gene expression in August 2019, February 2020, and February 2021, we found that the relationship between propolis score and immune gene expression differed among sample dates (Fig. 8). In August 2019, with the exception of AmEater, immune gene expression tended to decrease and stabilize with increasing propolis score. However, although abaecin expression tended to decrease with increasing propolis score in February of 2021 (Median: -0.066, 90% QCI: [-0.129, -0.004]), other genes demonstrated patterns of increasing gene expression and destabilization with increasing propolis score. Defensin-1 expression tended to increase with propolis score in February of 2020 (Median: 0.099, 90% QCI: [0.003, 0.199]) and 2021 (Median: 0.035, 66% QCI: [0.013, 0.053]) (Fig. 8). AmPPO expression tended to increase (Median: 0.164, 66% QCI: [0.049, 0.277]) and destabilize (Median: 0.141, 99% QCI: [0.022, 0.256]) in February of 2020, and tended to increase in February 2021 (Median: 0.037, 90% QCI: [0.024, 0.059]), and AmEater expression tended to increase (Median: 0.034, 95% QCI: [0.001, 0.066]) and destabilize (Median: 0.066, 66% QCI: [0.02, 0.111]) and in February of 2021.

L590: The expression of M. plutonius … change implemented

L595-597: If colonies were treated for Varroa, how does the reduction in infestation associate with the propolis collection? Thanks for pointing this out… We now see how the original language used was not clear. Varroa levels were measured prior to treatment. Changed sentence to: “Similarly, pre-treatment Varroa infestation (number of mites/100 bees) was lower in rough box colonies compared to control colonies by nearly one third, though this difference was non-significant (F(1,108) = 1.9, p = 0.18, Fig 9C).”

L598-599: It looks that changes in the viral load may correlated with mite levels. Please discuss it. Added: “We detected no significant correlation between viral loads and mite levels when both were tested in August 2019 (CBPV: r(105) = 0.05, p = 0.63; IAPV: r(105) = 0.13, p = 0.17; LSV-1: r(105) = 0.06, p = 0.53; LSV-2: r(105) = 0.04, p = 0.65; BQCV: (r(105) = 0.07, p = 0.49); ; ABPV: r(105) = 0.08, p = 0.42; DWV-A: r(105) = 0.001, p = 0.98; DWV-B: r(105) = 0.13, p = 0.17; KBV: r(105) = -0.001, p = 0.99” beginning on line 751, new version, and we added “The mechanism through which propolis impacts viral loads is unclear and could involve additional factors such as mite population (i.e., if propolis reduces mite loads, it may reduce viral transmission by extension). Future studies should examine the relationship between viral loads, mite levels, and propolis deposition across colonies with a wider range of Varroa infestation. In our study, the lack of correlation between Varroa mite levels and viral loads could be due to the low range in mite loads or it could be suggestive of a direct impact of propolis against these viruses.” To the discussion

L618-619: Explain why stationary colonies were wintered without enough food. The current sentence reads “High losses in the stationary yard were attributed to an issue with fall feeders, which prevented colonies from entering winter with sufficient food stores.” The issue was that the drip holes in the fall feeders – which were new that year – were too small, making it difficult for bees to drink from them. We are not sure this level of detail is necessary to include in the manuscript, but we are open to discussing further if necessary.

L626: Show statistic of results for honey production. Changed to: In the stationary colonies, there were no differences in mean honey production across treatments (Fig. 10; F(2,34) = 0.5, p = 0.6).

L637: Indicate the statistic for no effect of box type on honey production. Changed to: In year two, when colonies were started in boxes used in year one that were already propolized and when honey production was higher overall, there was no effect of box type on honey production (F(1,102) = 0.075, p = 0.8).

L669: Results did not support it. Results only compared rough boxes with control for migratory! This sentence actually refers to results from both the migratory and the stationary operation. It is true that, in the migratory operation, we only compared rough boxes and controls. However, in the stationary operation we compared rough boxes to controls and to propolis trap colonies. These results are further described in the sentences and paragraph that follow. Revised first sentence to further clarify: “Rough boxes were highly effective in stimulating propolis collection, compared to control boxes (stationary and migratory colonies) and boxes outfitted with propolis traps (stationary colonies).”

L673-676: Does difference between grooves sizes may make difference in propolis deposition? It might! But we tested boxes with uniformly sized groove sizes in this experiment so as not to add another variable.

L676: Overall, I suggest to show “mean± se/sd” for variables/amount, when you say more propolis deposited, more honey production or less survival. This helps readers to understand and follow the concept. This is a helpful observation. We went back through the manuscript and tried to include these values at all relevant points in the results section, but opted to exclude them from the discussion, to enhance readability.

L680: How much increase in the propolis deposition was measured over time? See L460-464 of original manuscript: “In August of 2019, propolis score averaged 3.2 (SE = 0.2) in rough box colonies, which was significantly higher than the 2.0 (SE = 0.2) average in control colonies (t(149) = 5.5; p < 0.0001). By February of 2021, propolis scores had more than doubled to an average of 7.2 (SE = 0.4) in rough box colonies (t(138.8) = 11.7; p < 0.0001) but remained stagnant at 2.2 (SE = 0.3) in control colonies.”

L691-692: Please explain the high or low plant biodiversity mean along with quantification/qualification and statistic. We’re unclear as to what this question/comment refers to. The sentence referenced is: “Previous research has established that areas of high plant biodiversity tend to provide more resin resources than areas of low plant diversity, corresponding to increased propolis production [40].” Is the reviewer requesting that we review the methods used in the publication we are referencing? 

L701: Could you give a statement on how different bee genotypes affect the results? See original submission: “Genetic differences may have also contributed to the variation we observed in propolis score, both between migratory and stationary yards, and between colonies in the same box type, in the same yard. Propensity for propolis collection is a highly heritable trait (coefficient of heritability = 0.87 [49]), and selection efforts can yield high-propolis colonies [50]. Different honey bee stocks were used in the stationary and migratory study, which, along with landscape differences, may have contributed to the variation in propolis deposition across contexts.”

L705: Is any genotype bee stock available in market for propolis production in the USA?

Not that the authors are aware of.

L712: Was a marginal reduction in Varroa load associated with propolis or miticide treatment? Miticide treatment was applied uniformly across box types (rough box vs. control) after quantifying varroa loads, but we now see how the original language we used was unclear. Added “compared to control migratory colonies” to clarify that the comparisons we are making are between the treatment groups we tested.

L714: 0.5 mites per 100 bees is low. How much is the seasonal mite threshold?

3 mphb is the current industry standard threshold, but it certainly can vary based on season (see O’Shea-Wheller et al. 2022, Sci Reports). This threshold can also vary by beekeeper and some bees definitely have variation in Varroa tolerance across stocks. We did not include this in the manuscript because we feel it is outside the scope of the current study.

L718: Is it meaning the propolis kills mites? In this study you did not measure the mite mortality, while mite mean abundance was evaluated. Pusceddu et al. found that the application of field-realistic quantities of propolis to artificial brood cells resulted in a near 20% increase in Varroa mortality. To avoid confusion, we changed the phrase “is consistent with” to “could be explained by.”

L722-726: This sentence is not relevant with the paragraph. Respectfully, we disagree. The preceding sentence states: “These effects have not been found to translate to reductions in mite loads at the colony level, possibly due to differences in the way researchers have attempted to simulate the propolis envelope.” The sentence on lines 722-726 then describes the specific differences in research methods - referenced in the previous sentence - which could result in contrasting effects on Varroa infestation. Finally, the final sentence in the paragraph reflects upon the possible different effects of a human-made propolis envelope (lines 722-726), versus a bee-made or propolis-trap-induced propolis envelope. 

L728-729: M. plutonius – Change implemented

L729: Delete “– the causative agent of European foulbrood (EFB)”. This was already cited. Change implemented

L730-731: To state like this, you should artificiality infect bees with pathogen, then see if propolis able to reduce the EFB infection in compared with the control. Changed to “The significantly lower levels in M. plutonius gene expression that we observed in bees from rough box colonies compared to bees from control colonies corresponded to marginally lower levels of colony signs of EFB in rough box colonies compared to control colonies the following month”

L736: The reduction in viral load may affected by other factors or interaction between biological aspect of bee colonies, mite population and propolis collection. Added “The mechanism through which propolis impacts viral loads is unclear and could involve additional factors such as mite population (i.e., if propolis reduces mite loads, it may reduce viral transmission by extension)…

L752: Propolis function can also be counted as an improvement in the immune system. True!

L755: “an increase in total bee …”. this may have related to bee genotype and environmental factors in compared with stationary. Here, it seems the original wording was not clear. We’re speaking only to differences in bee population in the migratory colonies, and comparing rough box colonies to control colonies (not migratory rough boxes to stationary rough boxes). Changed the sentence to read: Perhaps related to decreased pathogen pressure, in migratory colonies total bee population was higher in rough boxes colonies compared to control colonies at the end of year one of this study

L770-771: This study did not quantify the propolis. Quantification means that you measure the weigh of collected propolis. However, you evaluate propolis based on a scale not weight. Changed the word “quantified” to “used a numerical scoring system to determine”

L785-786: … with previous work (references). Added reference

L786: In migratory, propolis deposition had contrasting effects on …. Change implemented

L790-792: Different collected resin may affect the antimicrobial property of seasonal propolis. Thank you for bringing this up, as it is another hypothesis to explain this phenomenon. Added, “Additionally, since resins from different plants are known to have different antimicrobial properties, the seasonal differences we observed could result from seasonal differences in available resin resources [58].” 

L794: Did bees get any stress from agro-pesticides in the almond pollination time?

This is possible, but is outside of the scope of the current manuscript. No apparent issues were noted during colony assessments and or sampling. If agrochemical exposure did represent a significant stressor, colonies from both treatments would have been equally exposed. 

L813: 33 pounds per colony (= ??? kg/ colony) Thanks for catching this! We switched from pounds to kg per journal standards

L818: “a bad honey year”. Bad year in comparing with what? When we say beekeepers described year one as “a bad honey year” overall, we refer to beekeepers’ indications that honey production was lower during year one than it had been in previous years. These observations are also supported by our data (“Indeed, year one honey production was about half that of year two”).

In this study all colonies were compared with control. So if the year had a low honey season, it was equal for all treatments and control. That is correct. Here, we’re meaning to consider the possibility that lack of nectar availability could impact rough boxes and control boxes differently. (“It is possible that small rough box colonies produced less honey because foragers were occupied with resin collection, but since these colonies did not bring in significantly more propolis than small control colonies, there is no clear evidence indicating a nectar/resin tradeoff.”) If this point requires further clarification, we’re happy to add additional detail.

L828: Year one rough boxes were reused in year two with residue of propolis from last year, … changed to: “Since year one rough boxes were reused in year two, year two colonies were established in boxes containing propolis, and these colonies demonstrated no differences in honey production across treatments.”

---

## [Decision Letter · Decision Letter 1]

5 Sep 2023

Thinking inside the box: Restoring the propolis envelope facilitates honey bee social immunity

PONE-D-23-10474R1

Dear Dr. Shanahan,

We’re pleased to inform you that your manuscript has been judged scientifically suitable for publication and will be formally accepted for publication once it meets all outstanding technical requirements.

Kind regards,

Olav Rueppell

Academic Editor

PLOS ONE

Additional Editor Comments (optional):

Reviewers' comments:

Reviewer's Responses to Questions

**Comments to the Author**

1. If the authors have adequately addressed your comments raised in a previous round of review and you feel that this manuscript is now acceptable for publication, you may indicate that here to bypass the “Comments to the Author” section, enter your conflict of interest statement in the “Confidential to Editor” section, and submit your "Accept" recommendation.

Reviewer #2: All comments have been addressed

2. Is the manuscript technically sound, and do the data support the conclusions?

Reviewer #2: Yes

3. Has the statistical analysis been performed appropriately and rigorously? 

Reviewer #2: Yes

4. Have the authors made all data underlying the findings in their manuscript fully available?

Reviewer #2: Yes

5. Is the manuscript presented in an intelligible fashion and written in standard English?

Reviewer #2: Yes

6. Review Comments to the Author

Reviewer #2: (No Response)

7. PLOS authors have the option to publish the peer review history of their article (what does this mean?). If published, this will include your full peer review and any attached files.

Reviewer #2: No

---

## [Editor Report · Acceptance letter]

14 Sep 2023

PONE-D-23-10474R1 

Thinking inside the box: Restoring the propolis envelope facilitates honey bee social immunity 

Dear Dr. Shanahan:

I'm pleased to inform you that your manuscript has been deemed suitable for publication in PLOS ONE. Congratulations! Your manuscript is now with our production department. 

Kind regards, 

on behalf of

Dr. Olav Rueppell 

Academic Editor

PLOS ONE